# Monitoring, Reporting, and Verification for Ocean Alkalinity Enhancement

David T. Ho[1,2], Laurent Bopp[3], Jaime B. Palter[4], Matthew C. Long[2,5], Philip W. Boyd[6], Griet Neukermans[7], Lennart T. Bach[6]

[1]Department of Oceanography, University of Hawaiʻi at Mānoa, Honolulu, HI, 96822, USA
[2][C]Worthy, LLC, Boulder, CO 80302, USA
[3]LMD/IPSL, Ecole normale supérieure - PSL, CNRS, Sorbonne Université, Ecole Polytechnique, 75005 Paris, France
[4]Graduate School of Oceanography, University of Rhode Island, Narragansett, RI 02882, USA
[5]Oceanography Section, Climate and Global Dynamics Laboratory, National Center for Atmospheric Research, Boulder, CO, USA
[6]Institute for Marine and Antarctic Studies, University of Tasmania, Hobart, Tasmania, Australia
[7]MarSens Research Group, Department of Biology, Ghent University, 9000 Gent, Belgium

*Correspondence to*: David T. Ho (ho@hawaii.edu)

**Abstract.** Monitoring, reporting, and verification (MRV) refers to the multistep process of monitoring the amount of greenhouse gas removed by a carbon dioxide removal (CDR) activity and reporting the results of the monitoring to a third party. The third party then verifies the reporting of the results. While MRV is usually conducted in pursuit of certification in a voluntary or regulated CDR market, this chapter focuses on key recommendations for MRV relevant to ocean alkalinity enhancement (OAE) research. Early-stage MRV for OAE research may become the foundation on which markets are built. Therefore, such research carries a special obligation toward comprehensiveness, reproducibility, and transparency. Observational approaches during field trials should aim to quantify the delivery of alkalinity to seawater and monitor for secondary precipitation, biotic calcification, and other ecosystem changes that can feed back on sources or sinks of greenhouse gases where alkalinity is measurably elevated. Observations of resultant shifts in the partial pressure of $CO_2$ ($pCO_2$) and ocean pH can help determine the efficacy of OAE and are amenable to autonomous monitoring. However, because the ocean is turbulent and energetic and $CO_2$ equilibration between the ocean and atmosphere can take several months or longer, added alkalinity will be diluted to perturbation levels undetectable above background variability on timescales relevant for MRV. Therefore, comprehensive quantification of carbon removal via OAE will be impossible through observational methods alone, and numerical simulations will be required. The development of fit-for-purpose models, carefully validated against observational data, will be a critical part of MRV for OAE.

## 1 What is MRV?

In this chapter, we consider monitoring, reporting, and verification (MRV) for marine CDR (mCDR), confining our focus to determining the amount of additional $CO_2$ removed from the atmosphere and the durability of that removal. Investment in CDR is motivated by an interest in mitigating climate change, so the value of a CDR purchase stems from its correspondence to genuine removal (Smith et al., 2023). MRV must, therefore, provide estimates of net carbon removal and the uncertainty of those estimates (Palter et al., 2023). Delivering uncertainty estimates will enable markets to value carbon removal projects appropriately by applying discount factors scaled in accordance with uncertainty (Carbon Direct and Microsoft, 2023).

While we recognize the importance of determining ecosystem impacts of OAE deployments, assessment of OAE effects on ecosystems are covered in Eisaman et al. (2023, this Guide), Iglesias-Rodríguez et al. (2023, this Guide), Riebesell et al. (2023, this Guide), and Fennel et al. (2023, this Guide) and will not be considered MRV in this guide, unless they impact the efficiency of OAE (e.g., biogenic calcification). In addition to monitoring carbonate chemistry parameters for MRV (discussed below), assessing ecosystem impacts would require monitoring other biogeochemical, environmental, or ecological changes that may arise from OAE application, such as changes in nutrient fluxes, particulate loading, and phytoplankton community structure. In the same vein, side benefits (e.g., an increase in pH due to OAE) are also not considered MRV for this contribution. Finally, for this guide, we do not consider life cycle assessment (LCA), which might entail accounting for, e.g., $CO_2$ emissions from manufacturing, transportation, and deployment. While LCA is extremely important for quantifying the net carbon removed by a CDR strategy, this contribution focuses on MRV following OAE deployment in the ocean.

To determine the amount and duration of $CO_2$ removal, MRV must deliver an assessment of two interrelated metrics:

1. Additionality: The net quantity of $CO_2$ removal above a counterfactual baseline after OAE has been conducted in the ocean. Additionality should include assessments of phenomena such as precipitation-induced loss of alkalinity or a response in biogenic calcification that could reduce the ability of alkalinity addition to induce CDR.

2. Durability: The average time over which $CO_2$ is sequestered from the atmosphere by a given deployment. In our assessment, OAE minimizes concerns in the context of durability as OAE increases the ocean's buffer capacity and hence its ability to store $CO_2$ as dissolved inorganic carbon (DIC) on timescales associated with alkalinity cycling in the ocean— with residence time far exceeding $10^3$ years (Middelburg et al., 2020). Therefore, in our assessment, storage durability does not require an explicit methodology for quantification, but rather, we can assume that $CO_2$ removed via OAE will be stored mainly as bicarbonate ($HCO_3^-$) for $> 10^3$ years. For CDR, the depth of where atmospheric $CO_2$ is stored in the oceans matters when it is stored as dissolved $CO_2$ (as is the case for macroalgae cultivation or iron fertilization). However, in the case of OAE, $CO_2$ is stored mainly as $HCO_3^-$, which cannot be exchanged with the atmosphere, so surface ocean storage is chemically safe. Keeping alkalinity (and thus $HCO_3^-$) in the surface ocean has benefits for ocean acidification, although these are very minor and heavily depend on whether alkalinity-enhanced seawater has been equilibrated with atmospheric $CO_2$ (see Fig. 3 in Bach et al., 2019).

Furthermore, retaining alkalinity ($HCO_3^-$) in the surface ocean can enhance durability by limiting interactions with sediments and thus avoiding substantial loss terms to OAE, such as the risk of inducing secondary $CaCO_3$ precipitation in sediments and the reduction of natural alkalinity release (Fuhr et al., 2022; Moras et al., 2022; Bach, 2023; Hartmann et al., 2023). We acknowledge that there are also loss terms to alkalinity ($HCO_3^-$) in the surface ocean, such as the induction of biotic calcification. However, there is currently no reason to assume the deep ocean is a much safer place to store atmospheric $CO_2$ as $HCO_3^-$.

Further, as highlighted above, effective MRV systems must deliver estimates of the uncertainty in these metrics. To quantify these metrics, MRV for OAE must provide quantitative assessments in the context of the following questions:

1. How much alkalinity was effectively added to seawater? The difficulty of answering this question depends on the technology used for OAE. For example, understanding the dissolution kinetics of mineral particulates is a requirement to quantify alkalinity additions for crushed-rock feedstocks, but much less of a concern for electrochemical techniques and alkalinity added in dissolved form.

2. Has there been precipitation or biogenic feedback changing the efficacy of the alkalinity addition? Seawater is mostly above saturation in the surface ocean with respect to calcium carbonate; thus, the addition of alkalinity has the potential to induce precipitation of carbonate minerals (Moras et al., 2022), which would reduce the OAE efficiency (i.e., mole of DIC sequestered per mole of TA added). Abiotic $CaCO_3$ (or $MgCO_3$) precipitation is very slow but increases when the saturation state increases. Such high saturation states can occur near alkalinity release sites. Furthermore, calcifying organisms in the ocean, such as coccolithophores, can respond to OAE by modifying their growth rate or the relative amount of carbonate mineral production (Bach et al., 2019). Finally, enhanced saturation states could also reduce natural carbonate dissolution; this may have the effect of more effectively transferring alkalinity (in particulate form) from the surface ocean to depth or changing natural alkalinity sources from sediments or coastlines (Bach, 2023). Understanding these feedbacks of OAE via the calcium (magnesium) carbonate cycle is important for OAE MRV.

3. What is the ensuing perturbation to the air-sea exchange of $CO_2$ resulting from the OAE deployment? Alkalinity shifts carbonate equilibrium reactions away from aqueous $CO_2$, thereby reducing seawater $p\text{CO}_2$; CDR occurs when the atmosphere equilibrates with the altered surface ocean via air-sea $CO_2$ exchange. A primary goal for MRV is to quantify this perturbation flux; notably, however, in many envisioned circumstances, the alkalinity addition will be entrained in the ocean flow, causing the OAE signal to be transported away from the injection site and potentially away from the sea surface; coupled with the fact that $CO_2$ gas equilibration occurs slowly (Jones et al., 2014), the ensuing air-sea flux perturbation will occur over large regions in space and time.

In our assessment, observations alone are unlikely to provide a sufficient basis for quantifying the net carbon removal accomplished by OAE deployments. MRV for OAE requires the development of quantitative estimates of air-sea $CO_2$

exchange. Since the ocean is constantly moving and because $CO_2$ takes a long time to equilibrate across the air-sea interface, robust MRV would require intensive observations over large regions in space and time. High-quality carbon markets will require uncertainty bounds for net carbon removal estimates that would be prohibitively expensive to obtain via investment in direct observing over such scales, except, perhaps in targeted intensive observational arrays. A further complication with observations is that assessments of net carbon removals associated with OAE deployments require quantifying air-sea $CO_2$ flux relative to a counterfactual scenario: The air-sea $CO_2$ exchange that would have occurred without OAE intervention. Observing a counterfactual scenario is impossible in a strict sense, but it could be possible to use observations to assess counterfactual scenarios by leveraging analogs, such as nearby unperturbed regions, or statistical constructions, such as predicted seawater $p$$CO_2$ from empirical models built from historical observations of the carbon system and predictor variables like temperature, mixed layer depth, and chlorophyll (e.g., Landschützer et al., 2020; Rödenbeck et al., 2022; Sharp et al., 2022).

In practice, comparison with such analogs is a challenging task due to the heterogeneous nature of the ocean air-sea flux field, as well as the potential for OAE effects to spread over very large spatial and temporal scales. Notably, the background air-sea $CO_2$ flux field is highly dynamic on local to global scales. The ocean both absorbs and releases a massive amount of $CO_2$ each year; the net flux amounts to an uptake of about 10 Pg $CO_2$ yr$^{-1}$—but this net flux is a small residual of large gross fluxes (about ±330 Pg $CO_2$ yr$^{-1}$) (Friedlingstein et al., 2022). OAE can increase $CO_2$ flux into the ocean when the alkalinity enhancement reduces seawater $p$$CO_2$ below atmospheric $CO_2$. However, OAE can also decrease $CO_2$ flux into the atmosphere when alkalinity enhancement reduces seawater $p$$CO_2$ closer to atmospheric $p$$CO_2$. Both cases will constitute CDR as it leads to a net increase of DIC in the ocean reservoir (Bach et al., 2023). Geographic patterns of $CO_2$ ingassing and outgassing are controlled by the ocean's large-scale and subtropical overturning circulations (e.g., Iudicone et al., 2016), mesoscale and submesoscale motions (e.g., Nakano et al., 2011; Ford et al., 2023), variations in winds (e.g., Andersson et al., 2013; Nickford et al., 2022), storms (e.g., Nicholson et al., 2022), upwelling dynamics, local inputs from rivers (e.g., Mu et al., 2023), exchanges with sediments, and biology (e.g., Huang et al., 2023). Outside the tropics, there is pronounced seasonal variability in air-sea $CO_2$ fluxes mostly driven by phytoplankton blooms that draw down $CO_2$ in the surface ocean during spring and summer (e.g., Fassbender et al., 2022), and winter mixing that brings carbon-rich waters to the surface. All these dynamics are subject to variations in the climate and ocean circulation caused by internally fluctuating modes of variability or external forcing associated with $CO_2$ emissions and other human activities.

Given the complex nature of the ocean biogeochemical system, robust MRV for high-quality carbon removal markets will presumably depend on model-based approaches when quantifying net $CO_2$ removals. Ocean biogeochemical models (OBMs) will be a critical tool in this context (see Fennel et al., 2023, this Guide). These models represent the physical, chemical, and biological processes affecting the distribution of carbon, alkalinity, and nutrients in the ocean. OBMs represent inorganic and organic carbon pools, alkalinity, and nutrients as tracers with units of mass per volume (or mass) of seawater. OBMs are based on ocean general circulation models (OGCMs) that represent the movement of tracers mediated by ocean circulation and mixing. Biogeochemical tracers, including DIC and TA, have sources and sinks from processes such as

biologically mediated production and remineralization of organic matter. Boundary fluxes for OBM tracers include riverine
inputs, aeolian deposition, sediment-water exchange, and air-sea gas exchange. Fennel et al. (2023, this Guide) provide an
overview of the most relevant modeling tools for OAE research with high-level background information, illustrative examples,
and references to more in-depth methodological descriptions and further examples.

## 2. Specificities of MRV for marine CDR

The natural ocean carbon cycle is extremely dynamic on a wide range of temporal and spatial scales, typically
spanning more than ten orders of magnitude (Sarmiento and Gruber, 2006). These scales range from that of the ocean skin, a
thin layer of less than a millimeter in contact with the atmosphere where air-sea $CO_2$ exchange is controlled by molecular
diffusion, to that of the global ocean circulation that typically transports dissolved carbon over more than a thousand years and
10,000 km. As such, the ocean represents a challenging environment for MRV, especially compared to MRV of land-based
CDR techniques. Three specific time scales are to be considered when discussing challenges for MRV of mCDR, and in
particular OAE.
The first time scale relates to natural variability in carbonate chemistry, especially $p$CO$_2$ and alkalinity, due to
biological, chemical, and physical processes in the ocean. Such variability can be substantial on daily and seasonal time scales.
For example, using in situ observations from 37 stations spanning diverse ocean environments, Torres et al. (2021) showed
that in the open ocean stations, the average seasonal cycle of $p$CO$_2$ was $49 \pm 23$ µatm (inter-station mean and standard
deviation), and that diurnal variability could also be as high as $47 \pm 18$ µatm. Temporal variability at coastal stations where
OAE is likely to be deployed — due to proximity to existing infrastructure, energy supply, and human resources — is
significantly higher, with seasonal variability in $p$CO$_2$ being $210 \pm 76$ µatm and diurnal variability reaching $178 \pm 82$ µatm
(Torres et al., 2021). OAE-induced changes in $p$CO$_2$ are likely to be lower than the range in natural variability, complicating
MRV. For example, an increase in alkalinity of 10 µmol kg$^{-1}$ would result in a decrease in $p$CO$_2$ of around 20 µatm (given
temp $=20°C$; salinity $= 35$; initial TA $= 2200$ µmol kg$^{-1}$; DIC $= 1965$ µmol kg$^{-1}$ and no secondary precipitation or biotic
calcification). Historical carbonate system variability, like the examples given here, can be used in sensitivity studies to assess
the detectability of a given OAE perturbation for different observing systems (Mu et al., 2023).
The second of these time scales relates to air-sea $CO_2$ equilibrium. This time scale is particularly relevant for OAE as
it determines the time required from an alkalinity-driven shift in surface seawater carbonate equilibria to a new air-sea $CO_2$
equilibrium and the resulting atmospheric carbon uptake. It is well established that the characteristic timescale for air-sea
exchange of $CO_2$ is of the order of 6 months (Sarmiento and Gruber, 2006). But Jones et al. (2014) have shown that the time
to reach air-sea $CO_2$ equilibrium is highly variable at the regional scale, ranging from less than a month to several years, with
especially long values in the northern North Atlantic, the Atlantic subtropical gyres, and the Southern Ocean. This regional
variability is explained by the dependency of the air-sea $CO_2$ equilibrium time scale on the gas transfer velocity, the depth of
the mixed layer, and the baseline carbonate chemistry of seawater. More precisely, this time scale shortens with higher gas
transfer velocities and Revelle factors, but lengthens with deeper mixed layers and larger ionization fractions (i.e., the ratio
between DIC and dissolved $CO_2$).
The third of these time scales relates to ocean physical processes and alkalinity and carbon transport away from the
injection location. First, horizontal currents, ranging from a few centimeters to a few meters per second, can potentially
transport the OAE signal away from the initial injection site, thus complicating MRV. A simple calculation shows that a mean
flow of 0.5 m s$^{-1}$ could transport the alkalinity signal more than 100 km from the initial site in six months. Second, vertical
entrainment, mixing, and/or other subduction processes might also transport the OAE signal to depths below the seasonal
mixed layer, potentially hindering atmospheric $CO_2$ uptake and associated MRV.
Lessons learned from mesoscale in situ ocean iron fertilization (OIF) studies can be applied to MRV for OAE,
especially during pilot studies of unenclosed OAE-perturbed patches of surface waters that are upscaled beyond a few km$^2$.
Ocean circulation and mixing will cause a range of effects that are scale-dependent and will influence MRV strategies as it is
used to target pilot studies and, eventually, larger deployments (100 km$^2$ scale). This presupposes that elements of MRV will
be needed at all spatial scales during the development and testing of an mCDR method.
The success of OIF in tracking and the repeated sampling of a coherent patch of perturbed waters over a timescale of
weeks was due to the use of $SF_6$ as an ocean tracer (e.g., Coale et al., 1996), and, in one instance, using a quasi-controlled
volume (e.g., within a mesoscale eddy; Smetacek et al., 2012). For example, the use of $SF_6$ allowed dynamic upper ocean
behavior to be observed during an OIF perturbation, in which the perturbed water was subducted under less dense water in a
few days, leading to the termination of the study (Coale et al., 1998). Subduction is a risk for the MRV of OAE trials being
conducted in nearshore waters, and the use of tracers such as $SF_6$ would be crucial for observing this behavior.
At larger spatial scales (i.e., for perturbations done in waters not bounded by eddies >100 km$^2$), ocean physics imposes
a strain and concurrent rotation of a perturbed patch of ocean; as such, OIF studies revealed the perturbed patch of waters can
'grow' in areal extent from 100 km$^2$ to > 1000 km$^2$ via the entrainment of the surrounding 'control' seawater (Law et al., 2006).
Such entrainment sets up concentration gradients that lead to fluxes into (in the case of OIF, nutrients are resupplied to the
nutrient-depleted patch) and out of (in the case of OIF, chlorophyll which has accumulated due to OIF, and iron that has been
added) the perturbed waters. Such artifacts may dilute the more alkaline waters in the patch of unenclosed OAE perturbed
waters, which may hinder aspects of MRV such as detection of the OAE signal above a background level, or biological side-
effects resulting from OAE.

## 3. Observation-based techniques for MRV and limitations

OAE depends on multi-step processes to achieve mCDR: First, the intervention raises ocean alkalinity in order to
lower seawater $p$$CO_2$, and then atmospheric $CO_2$ must equilibrate with the altered waters. These processes point to many of
the variables that would ideally be observed in an OAE MRV scheme. Measurements of total alkalinity (TA) and DIC are
important to quantify the background state of the carbon system, which determines the $p$$CO_2$ response per unit change in

alkalinity. Further, measurements of TA might help verify that alkalinity has been added effectively, although signal-to-noise ratios may be insufficiently strong to enable robust detection and attribution of TA anomalies (Mu et al., 2023). pH is an important measurement to ensure that the OAE deployment conforms with water quality limits (usually pH < 9) and that the deployment does not create conditions that induce precipitation. Finally, $pCO_2$ at the ocean's surface is a key control on gas exchange and is thus an important measurement target. With extensive measurements of these variables along the Lagrangian pathway of a perturbed water mass, a carbon budget could theoretically be closed by constraining the time-rate of change and making inferences about important driving processes such as air-sea gas exchange; such a budget could, in theory, be used to support quantification of CDR for a given OAE deployment. Though appealing in its comprehensiveness, the reality of observing all of the parameters needed to quantitatively close a perturbed carbon budget and compare it against an unperturbed counterfactual is likely impossible in the near to medium-term, even in the context of highly-monitored field trials. The difficulty is inherent in the fact that the patch of water perturbed by the addition of TA is likely to be turbulently dispersed in the ocean, and its signal diluted below the limit of detectability by mixing over the time scale required for $CO_2$ equilibration (He and Tyka, 2023; Mu et al., 2023; Wang et al., 2023).

This leads to the conclusion that MRV via direct observational approaches should not be expected to completely follow every molecule of additional $CO_2$ resulting from an OAE deployment - as doing so would set an insurmountable barrier to MRV. Instead, we outline what can feasibly be observed, what questions these observations can answer, and which questions are left to be addressed in statistical and/or prognostic models with their attendant uncertainties.

Various autonomous sensors hold promise to inform the results of an OAE deployment, both in field trials and for sampling that might offer constraints on open water applications and data for model validation and/or assimilation.

The most direct measurement relevant to OAE experiments is TA, which would reveal if the initially planned perturbation was successful. Though autonomous sensors for TA have been in development for several years (Briggs et al., 2017), they are not commercially available at the time of writing, and the laboratory analysis of bottle samples cannot currently be replaced or even supplemented by sensor-based measurements (see Cyronak et al., 2023, this Guide). Nevertheless, laboratory analysis of TA in bottle samples can be compared to "baseline" measurements taken before the alkalinity is added or outside the expected patch area. It is worth noting that measuring a TA increase near the OAE deployment point may be possible, but once the OAE-perturbed water has dispersed in the ocean flow, the signal-to-noise ratio will likely be too low to make any accurate quantification. This is also the case for attempting to quantify CDR using DIC, as discussed below. The TA in the OAE-influenced patch may also be compared to a predicted counterfactual TA constructed from regression methods built with historical salinity (and other available) data, like the Locally Interpolated Alkalinity Regression (LIAR) method (Carter et al., 2018).

In contrast to TA, to determine the ocean uptake of $CO_2$, there are effective equilibrator-based autonomous $pCO_2$ systems (e.g., ASVCO2™, MAPCO2) capable of measuring $pCO_2$ with a nominal accuracy of 2 $\mu$atm (R. Wanninkhof, Personal Communication), although they are restricted to the top few meters of the surface ocean due to the fact that equilibrators cannot be submerged. There are also in situ $pCO_2$ sensors that rely on equilibrating seawater $pCO_2$ with air

through a membrane (e.g., Pro-Oceanus CO2-Pro™ CV, CONTROS HydroC® CO2) or a pH-sensitive dye (e.g., SAMI-pH), followed by infrared detection or colorimetric spectroscopy. Due to fluctuations in the pressure of equilibration and calibration issues, the real-world accuracy of these instruments is ~5 µatm (R. Wanninkhof, Personal Communication). The existence of autonomous $pCO_2$ sensors is potentially important because while it is difficult to detect changes in the carbon inventory of the ocean with measurements of DIC, it can be done with measurements of $pCO_2$ (Wanninkhof et al., 2013). These $pCO_2$ sensors can be deployed on moorings (MAPCO2, ProCV) and autonomous surface vehicles like Wave Glider (ASVCO2) (Chavez et al., 2018) and Saildrone (Sabine et al., 2020; Sutton et al., 2021; Nickford et al., 2022). These sensors have the advantage of being able to collect measurements continuously in harsh weather and with much reduced involvement from skilled analysts relative to field surveys with bottle collection. Most analysis focuses on collecting and analyzing calibration samples and performing quality control on data.

Sensors that measure pH on autonomous profiling floats, gliders, or moored platforms could provide additional data useful for MRV. Unfortunately, as demonstrated by Wimart-Rousseau et al. (2023), pH sensors on profiling floats have relatively large uncertainties that may compromise their usefulness for MRV. Moreover, these uncertainties are largest near the ocean's surface, where they would be most useful in the MRV context, as knowledge of the surface ocean disequilibrium is needed for CDR. Uncertainties in pH of 0.01 roughly translate to a $pCO_2$ uncertainty of 10 µatm (Wimart-Rousseau et al., 2023), but even achieving such accurate pH measurements will require significant advances in sensor accuracy and/or post-processing data analysis tools to correct surface pH data.

Another MRV-relevant aspect of OAE that is well suited for sensor measurements is the reduction of OAE efficiency via OAE-induced precipitation of carbonates (see Schulz et al., 2023 for further context). For example, marine calcifiers, such as coccolithophores, may benefit from high alkalinity and pH conditions, thus reducing OAE efficiency (Bach et al., 2019), but this effect is still uncertain (Gately et al., 2023). Autonomous optical sensors for particulate inorganic carbon (PIC) based on the birefringence of calcite and aragonite have been in development for several decades (James, 2009; Bishop et al., 2022). Since the deployment of the first prototype on a profiling float in 2003, this optical PIC sensor has been re-engineered several times, and the most recent versions require further re-engineering to correct for thermal and pressure effects, as well as misalignment effects of the linear polarizers (Bishop et al., 2022). A new autonomous PIC measurement concept was recently proposed by Neukermans and Fournier (2022), which may overcome the aforementioned issues. Such PIC sensors are currently under development and are expected to cover a PIC concentration range of 0.5 to 500 µgC $L^{-1}$ (Neukermans et al., 2023). These PIC sensors are intended for use on autonomous platforms such as floats profiling up to 2000 m deep, autonomous moorings, tethered buoys, or Saildrones. Such PIC sensors would thus enable careful autonomous monitoring of PIC concentration in the epi- and mesopelagic ocean, as well as in shallow shelf seas. In addition, ocean color satellites can be used to obtain global maps of coccolithophore PIC concentration in the surface ocean at daily frequency using a variety of remote sensing algorithms (see Balch and Mitchell, 2023 for a review of remote sensing PIC algorithms and limitations). Both remote sensing and in situ observations of PIC concentration can contribute to assessing secondary precipitation and OAE efficiency.

60        Other more remote tail risks of OAE include alterations to carbon production and flux, for example, via shifts in

phytoplankton community structure (Ferderer et al., 2022) or alterations in the availability of high-density biominerals such as
opal or calcite, which may ballast POC flux to the deep ocean (Armstrong et al., 2001; Klaas and Archer, 2002). Ballasting of
POC flux by coccolithophore calcite and the resulting increase in the sinking velocity of POC aggregates has been confirmed
in many experimental studies and may be an important mechanism in some ocean regions. This potential secondary effect of
OAE on POC flux could be monitored from autonomous profiling floats equipped with a PIC sensor (Neukermans et al., 2023).

66        Wind speed should be measured since it is the most common correlate for air-sea gas exchange, and there are wind

speed/gas exchange parameterizations that predict gas transfer velocities well in the open ocean (e.g., Ho et al., 2006).
Therefore, in these settings, measurements of wind speeds are sufficient to characterize air-sea gas exchange. However, since
gas transfer velocities as a function of wind speed differ between the open and coastal oceans (e.g., Dobashi and Ho, 2023),
depending on the OAE deployment location, $^3$He/SF$_6$ tracer release experiments might have to be performed to determine this
relationship (see Wanninkhof et al., 1993). While it is likely unfeasible to couple every individual OAE operation with a
$^3$He/SF$_6$ dual tracer release during the deployment phase, during the testing phase, such experiments will be useful for
calibrating and evaluating models that will most likely be used to determine the efficiency and efficacy of CO$_2$ equilibration.

## 4. Model-based techniques for MRV and limitations

75        OBMs can be used to explicitly represent the effects of OAE by conducting numerical experiments in which the

model is provided with forcing data that represents alkalinity additions. Developing and validating models in the region/scale
of OAE deployment should be a priority to enable functional frameworks for MRV (see Fennel et al., 2023, this Guide).

78        A model integrated forward in time with the alkalinity additions will simulate the transport of the associated mass of

alkalinity and its ensuing effect on biogeochemical processes, including air-sea gas exchange. These simulations can be used
to evaluate net carbon removal by comparing integrations that include the OAE signal to others in which that forcing is not
present — i.e., the baseline counterfactual condition or "control." If an ensemble of integrations is performed, the variation of
net carbon removal across the ensemble can be used to assess uncertainty. Notably, there are different potential sources of
uncertainty: If intrinsic variability in ocean dynamics is considered the dominant source of uncertainty, an initial condition
ensemble could provide an appropriate representation of uncertainty. If model structure, in contrast, is the dominant source of
uncertainty, alternative approaches to ensemble construction could be employed, including perturbing parameters or using
multiple models (see Fennel et al., 2023, this Guide for further discussion). Explicit simulation of OAE deployments can be
compared to observations, including measurements from background observing systems, as well as bespoke data collection
efforts associated with the OAE project. In some cases, explicit data assimilation (DA) procedures may be applied (see Fennel
et al., 2023, this Guide), potentially reducing model-data misfits and improving confidence in the model simulations. One
challenge of applying DA to MRV is estimating additionality, which requires information about both the actual temporal
evolution of the system and the counterfactual condition, i.e., the state of the system that would have occurred in the absence

of the CDR intervention. The counterfactual condition is impossible to observe directly, and to the extent that observations contain an imprint of the CDR, DA cannot be used to generate explicit estimates of the baseline state. This raises conceptual issues because simulations conducted with and without DA are not directly comparable; thus, a difference between DA-constrained and free-running models cannot provide a valid estimate of additionality. Further research is needed to understand and address these problems. Potential solutions may rely on the assumption that CDR signals are very small relative to the background variability and, thus, essentially negligible in the context of the constraints on model solutions imposed by DA. Further, if the CDR interventions can be assumed to have negligible impact on physical variables (e.g., temperature, salinity, currents, etc.), it may be possible to use DA selectively on just these variables.

## 4.1 Modelling alkalinity addition

For the effects of OAE to be properly simulated, models must be supplied with the correct amount of alkalinity applied as forcing. Alkalinity additions, if performed over hours to days, are likely to occur on scales much smaller than the ensuing anomaly generated in air-sea $CO_2$ exchange, typically occurring over months to years (see Section 2). For this reason, MRV frameworks must invoke a separation of concerns, wherein near-field (i.e., within a few km of the source) processes are treated differently than the broader regional effects. Explicit modeling of near-field dynamics is likely to require different modeling frameworks (e.g., McGillicuddy, 2016) than those simulating the full expression of the OAE effects in the ocean—however, it is not necessarily a requirement to simulate near-field dynamics in the context of MRV. Near-field processes must be constrained by direct observations, and/or their dynamics must be accurately captured in verified parameterizations applied to models too coarse to simulate the local effects explicitly (e.g., Fox-Kemper et al., 2019). Notably, different OAE technologies and feedstocks present different challenges in this regard (see Eisaman et al., 2023, this Guide). Electrochemical techniques, which might produce, for instance, an alkalinity-enhanced stream from an outfall pipe, are different from crushed-rock particulates where dissolution kinetics come into play. Moreover, as discussed in Fennel et al. (2023, this Guide), ancillary constituents (e.g., iron or nickel) associated with rock-derived feedstocks may induce biological responses with impacts on the total efficacy of the OAE process.

## 4.2 Representing OAE effects

To provide a suitable basis for MRV applied to OAE deployments, models must meet several requirements and provide a sufficiently accurate representation of alkalinity additions. First, models must provide a reasonable representation of ocean circulation and mixing; these processes are critical to determining the residence time of added alkalinity in the surface mixed layer, where gas exchange with the atmosphere is possible. Given that the equilibration time scale for $CO_2$ via gas exchange is long, the residence time of alkalinity-enhanced water parcels at the ocean surface is likely a primary control on the efficiency of uptake (He and Tyka, 2023). Second, the models must accurately capture the surface ocean $p$CO$_2$ anomaly induced by alkalinity additions. This implies having a correct representation of the carbon system thermodynamics (see Fennel et al., 2023, this Guide). Further, since the change in $p$CO$_2$ depends on the background DIC:TA ratio (Hinrichs et al., 2023), it

is important that the model has a good representation of the mean state prior to perturbation (Planchat et al., 2023). Third,
presuming an accurate representation of the change in $p\text{CO}_2$ and the transport of alkalinity following injection, the model must
be able to simulate the gas transfer of $\text{CO}_2$ with sufficient accuracy. Notably, the gas transfer velocity is highly uncertain,
particularly in coastal environments where many OAE deployments are likely to occur (e.g., Dobashi and Ho, 2023). If surface
water residence times are much longer than the gas equilibration timescale, uncertainty in the gas transfer velocity may not
contribute substantially to the overall uncertainty—but in intermediate regimes where the two timescales are comparable,
uncertainty in the gas transfer velocity may be an important consideration. Finally, a comprehensive assessment of OAE
efficacy will depend on accurate characterization of feedbacks in the biological system. If there are changes in the natural
distribution of calcification or organic carbon export, this term should be quantified—or its potential magnitude and impact on
overall carbon transfer should be assessed as a component of the uncertainty budget. At present, further empirical research is
required to enable modeling systems to treat this aspect of OAE effects robustly (Fennel et al., 2023, this Guide).

## 5. The way forward for MRV of OAE

There is much work to be done to establish how to optimize monitoring OAE with respect to which observations are
needed and at what spatial and temporal resolution and duration. Nevertheless, early field trials should all monitor the initial
increase in alkalinity (i.e., both measured and modeled). Baseline alkalinity measurements should be made so that the range
of concentration within its natural variability is known before the deployment of alkalinity. Furthermore, if the enhancement
is done via the dissolution of pulverized rocks, the dissolution rate needs to be known under in situ conditions. Knowledge of
this rate includes the dependency on various factors such as temperature, salinity, etc. but also to what extent minerals become
buried in sediments and how this change in exposure affects dissolution. If the enhancement is done via electrochemistry, the
dosing rate of the solution (e.g., $\text{Mg(OH)}_2$, NaOH) should be quantified and reported with complete information about the
measurement methods and a thorough accounting of their uncertainties.
Furthermore, any potential secondary precipitation caused by the alkalinity enhancement (e.g., if alkalinity is added
too quickly, brucite precipitation could occur) should be monitored. Monitoring of secondary precipitation is particularly
critical in the non-equilibrated state (i.e., before atmospheric $\text{CO}_2$ influx has occurred) and when the alkalinity-perturbed patch
is in close contact with sediments since the risk for secondary precipitation is particularly high under these circumstances (see
Eisaman et al., 2023, this Guide; Schulz et al., 2023, this Guide).
Finally, the drawdown of $\text{CO}_2$ in the ocean due to alkalinity addition should be measured. Given the potential natural
variability in $p\text{CO}_2$, especially in coastal regions, monitoring of $p\text{CO}_2$ should also be done before the OAE deployment.
Considering the spatial and time scales discussed above, these measurements will need to be complemented by modeling
approaches.
MRV of $\text{CO}_2$ influx after the application of OAE will likely depend on fit-for-purpose modeling (see Fennel et al.,
2023, this Guide). Exceptions to this may apply if the deployment is made in an enclosed area where the water is confined, or
the deployment is made in a heavily instrumented and surveyed area of the ocean. Models used to constrain atmospheric $CO_2$
influx must be calibrated and evaluated with observations. Since $CO_2$ influx is due to physical and chemical processes, the
following observational data to improve the modeling framework includes (but is not restricted to):

59            ●   Observations of ocean currents from acoustic Doppler current profilers (ADCPs), Lagrangian floats or tracers like

60               $SF_6$, and remote sensing;

61            ●   Observations of air-sea gas exchange from $^3He/SF_6$ tracer release experiments;

62            ●   Temperature and salinity profile measurements;

63            ●   Measurements of carbonate chemistry parameters (i.e., TA, pH, $pCO_2$, and DIC).

While it appears that OBMs will ultimately provide a critical foundation for robust ocean MRV frameworks, they are
not currently ready to serve in this capacity (Fennel et al., 2023, this Guide). These models represent complicated systems;
Ocean General Circulation Models (OGCMs) are based on fundamental governing equations, but solving these equations
numerically requires approximations (e.g., Fox-Kemper et al. 2019). Ocean ecosystems comprise diverse groups of organisms
with differing physiological capacities and complex interactions. There are no generally accepted governing equations for
these systems; rather, models are built on the basis of empirically determined relationships and theory or hypothesis (e.g.,
Planchat et al., 2023). For OBMs to provide a credible basis to support ocean MRV, they must be based on broadly accepted
theory or well-constrained parameterizations, and they must be explicitly validated relative to the quantification of gas
exchange anomalies arising as a result of perturbations in alkalinity. Models have not yet been robustly validated in the context
of these explicit requirements.
We note that at this point, we have yet to develop the best modeling tools for OAE MRV (and likely MRV for mCDR
in general). A rigorous research and development program to establish OBMs as fit-for-purpose, credible tools for MRV are
needed. However, there is currently a major problem with basing MRV on models. OBMs are run on high-performance
computing architectures, and because they are big calculations, they are very computationally expensive (and therefore
financially expensive). It is unlikely that technological innovation will dramatically reduce this computational cost in the next
5-10 years, during which time we will be required to deliver a functional framework for MRV. Therefore, we suggest
combining direct model simulations with advanced statistical approaches to overcome the computational challenges. First, we
must establish that models can provide credible representations of key CDR processes by ensuring that model output agrees
with available observations. Then, we can leverage these models to generate datasets from which to derive robust statistical
approximations, including through the application of techniques derived from artificial intelligence and machine learning. For
instance, well-calibrated models could be used to produce training data for machine learning algorithms to predict the CDR
efficiency of OAE deployments in different locations at different times, i.e., as a function of initial environmental conditions
such as water temperature, carbonate chemistry, mixed layer depth such as suggested in Bach et al. (2023).
Conducting explicit OAE modeling experiments coupled with field trials are important research milestones necessary
to identify the long-term approach to robust MRV. It is likely that the models that can effectively support field trials will use
regional OGCMs that are capable of high-fidelity simulations of ocean flows at scales commensurate with those driving the

initial dispersion of OAE signal on timescales of weeks to months. Unless alkalinity is continuously applied at a level measurable by long-duration observing platforms, the OAE signals are likely to be diluted and less easily tracked with observations. Critically, it is important to demonstrate that the models provide simulations consistent with the carbonate chemistry and deliberate tracer observations.

Models that compare well to observations can be deemed credible for assessing OAE effects. However, fully explicit mechanistic calculations are computationally intensive and thus unlikely to provide a scalable framework for conducting MRV under the scenario of widespread OAE deployments. On this basis, it is important that research on OAE field trials aims toward building trust in models to develop approaches to MRV that can be accomplished at a reduced computational cost.

## 6. Key recommendations for MRV of OAE

Early-stage MRV research for OAE may become the foundation on which regulated markets are built. Therefore, such research carries a special obligation toward comprehensiveness, reproducibility, and transparency. To fulfill these obligations, we suggest the following overarching best practice guidelines:

- Field trials should be co-designed with modelers and observationalists to enable the iterative process of model validation and improvement and dynamically informed data interpretation. In some scenarios, co-design may entail the development of formal Observing System Simulation Experiments, and data-assimilating state estimates (Fennel et al., 2023, this Guide).

- MRV techniques and results should be well-documented and archived publicly and promptly, without restriction (e.g., Planetary Technologies, 2023). Ideally, a central registry of OAE experiments would adhere to FAIR (Findable, Accessible, Interoperable, and Reproducible) data standards (Wilkinson et al., 2016). Researchers should eschew any practice that withholds MRV innovation from the community to "build a moat" in support of a commercial mCDR approach.

- Early field trials are recommended to be as comprehensive as possible, monitoring for obvious, first-order risks like secondary precipitation and more remote tail risks like alterations to export production via shifts in phytoplankton community structure and mineral ballasting.

- Model evaluation against observations should be tailored to the key processes in question. Fennel et al. (2023, this Guide) argue that models may be used for a long list of purposes, including, for example, simulating ecosystem effects and sediment-water exchanges. Early MRV efforts can expose model skill and deficiencies in simulating these processes if the relevant observations are prioritized.

- An uncertainty budget should be quantified that includes both known uncertainties (e.g., measurement and mapping errors) and expert estimates of presently unmeasurable risks. A comprehensive assessment of the poorly constrained uncertainties will point to key research areas in the future.

**Competing interests**

DTH and MCL are Co-Founders as well as Director of Science and Executive Director, respectively, of [C]Worthy, LLC, a non-profit research organization focused on building open-source tools to support MRV for marine CDR. DTH is also a Science Advisor at Carbon Direct, Inc., an end-to-end carbon management company. LTB is a scientific advisor to Submarine, a start-up service provider for MRV of marine CDR.

**Acknowledgments**

This is a contribution to the "Guide for Best Practices on Ocean Alkalinity Enhancement Research." We thank our funders the ClimateWorks Foundation and the Prince Albert II of Monaco Foundation. PWB was supported by the Australian Research Council (ARC) through a Laureate (FL160100131). LTB was supported by the ARC through Future Fellowship (FT200100846) and by the Carbon-to-Sea Initiative. GN has received funding from the European Research Council (ERC) under the European Union's Horizon 2020 research and innovation programme (Grant agreement No. 853516 CarbOcean), UGent's Industrial Research Fund (F2020/IOF-StarTT/088), and Special Research Fund (BOF/STA/202002/011). Thanks are also due to the Villefranche Oceanographic Laboratory for supporting the lead authors' meeting in January 2023.

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
