# Peer review of "Monitoring, Reporting, and Verification for Ocean Alkalinity"

_State of the Planet, 2023_

## Referee Comment (RC2)

Review of Ho et al. : Chapter 6: Monitoring, Reporting and Verification for Ocean Alkalinity Enhancement

This paper is a contribution to a "Guide to Best Practices in OAE research", which is a timely and highly relevant endeavour. The chapter itself is rather short, maybe because it is sandwiched between the Fennel et al modelling chapter and the Schulz et al. chapter on carbonate chemistry measurements. Overall, the chapter is well readable and informative.

Besides a few specific comments, my major comment is on incomplete referencing. The full first section ("1. What is MRV") only includes one reference (plus two references to other chapters in this issue, which are, however, missing in the reference list). Even though I agree with most statements, the full chapter reads more like an opinion piece rather than a scientifically sound paper deeply rooted in the scientific literature. Given that the key message by the authors is that "early stage MRV research for OAE [...] carries a special obligation toward comprehensiveness, reproducibility, and transparency" (Abstract and section 5), I urge the authors to follow their own recommendation and to thoroughly reference the relevant literature. The links and references may be obvious to those deeply involved in this research, but this series of chapters should have the ambition to be a starting point to those new in the field. Thus, please point the reader to relevant literature, so that he/she/it can find further reading material. I give a few examples below, but this holds for the entire manuscript.
My first thought was that there might be a common decision/recommendation of a maximum number of references to be included in the chapters of the best practise guide. If this was the case, I would strongly urge the entire group to rethink their decision. However, looking at the Fennel et al. paper with a substantially longer (though maybe also not complete) reference list, this does not seem to be the case.

In the same spirit, the manuscript would benefit from sometimes sparring a few more words for an additional explanation when statements are rather vague and unspecific, examples are given below.

Specific comments:

Line 20: 'pCO2 and pH' : pH is not mentioned at all in section 3 on measurements. Please add.

Line 28-29 (and in general): MRV is defined to deliver 'the amount of additional CO2 removed from the atmosphere' and 'the durability of that removal'.
What I am missing in this paper is twofold:
- any ocean-centric MRV being through observations or modelling would overestimate the atmospheric CO2 draw-down by neglecting feedbacks from atmosphere and particularly land and thus overestimating atmospheric CO2 reduction by ~25% on decadal scales (Oschlies, 2009, www.biogeosciences.net/6/1603/2009/ ). There may be ways to account for this even if not using an emission-driven Earth System Model, but these approaches need to be developed, and awareness needs to be raised. I think this is mentioned in the Fennel et al chapter, but don't count on everyone reading all chapters.

- I think it is too simplistic to say that all carbon in the ocean counts equally as all alkalinity stays in the ocean for long time-scales. Isn't it still better (for humans) to get the carbon out of the surface mixed layer and ideally into water masses that will not be in contact with the atmosphere for centuries? I.e., what about the depth of carbon sequestration? Shouldn't this be included in MRV where the carbon goes (horizontally and vertically)?

Line 36: you here promise the reader some text on methane, N2O and DMS. I did not find this mentioned again... add text (preferred) or delete this in our outline.

Line 45: "CO2 that escapes removal": there will be a lot of CO2 that will not be removed, so should this be reformulated to be more specific?

Line 47: "such as": just wondering whether there is anything more to add to the list or if it is complete then drop "such as".

Line 52: The long time-scale of alkalinity cycling in the ocean of Middleburg was challenged by Köhler (2023), who found, for example, a glacial-interglacial amplitude in alkalinity of 100 µmol/kg, which, according to the paper, is equal to 100 ppm in atmospheric CO2. This may not be relevant to the timescales of CDR, but some caution/toning down might be worth considering. I am not sure we have a sufficiently good understanding of changes in river/erosion input of organic and inorganic matter to the coastal ocean and neither of how sedimentary processes in coastal regions change with climate change and other human impacts to be so certain that alkalinity cannot be lost (other than through secondary precipitation and calcification).

Line 64: "stored as CaCO3": CaCO3 formation is usually thought to increase ocean pCO2, so the formulation "CO2 stored as CaCO3" seems to be misleading, and might need a few extra words.

Lines 64-70, 73-74, 76-78, 81, 85-86, 91-93!, 95-100, 114-116, 137: essentially, each sentence needs a reference.

Line 82: one does wonder: there are a couple of global pCO2-products estimating the ocean carbon sink from pCO2-observations. How does MRV relate to these estimates? Do you also consider them unreliable or what makes the difference to quantifying air-sea CO2 flux after OAE from observations? Is it the small scales? It would be welcome, if you commented on that for the reader.

Line 89: hmm, if a long-term climatology from observations would be used as a 'counterfactual scenario', what if the climate state/weather would be so different in the year of OAE (climate variability) that the different climate state/weather can explain all the differences in CO2 uptake. Should a company get credit for this?

Section "2: Specificities of ocean CDR for MRV": it would be nice to embed this for the reader in how this differs from land MRV (just a thought).

Line 122: 'coastal stations, where OAE is likely to be deployed': needs a ref or explanation

Line 134: "ionization fraction": is this not a very similar measure to the Revelle factor?

Line 136ff: not only alkalinity, also carbon will be transported (is that included in your 'OAE signal'?)

Line 142ff: "Lessons learned from OIF": it would be useful to extend this part and explain better. Currently, I do not take any lesson home from this text.

Line 147: "in the context of M (Measurement)": I thought M is Monitoring, but also I don't get the meaning of this phrase.

Line 156-158: not true, at least not always, see Smetacek et al 2012: "A large diatom bloom peaked in the fourth week after fertilization. This was followed by mass mortality of several diatom species that formed rapidly sinking, mucilaginous aggregates of entangled cells and chains."

Line 159: I'm not sure these are 'experimental artefacts' and 'do not represent the ocean C cycle', but yes, even well-defined experiments take place in a complex Earth System and have limitations. The experiments took place in the real world though and do represent the carbon cycle, though maybe not in its entirety or the aspects that you implicitly refer to (please specify).

Line 160: please explain why OAE would be less impacted by physical effects. Alkalinity and carbon are subject to circulation and mixing, only they may not be subject to the same potentially large amount of biological feedbacks.

Section 3: Observation-based techniques:
- as the abstract mentions pH, some text on pH sensors and their limitations should be added. I warmly recommend Wimart-Rousseau et al (BGD): "In the context of converting surface ocean pH measurements into pCO2 data for the purpose to derive air-sea CO2 fluxes, we conclude that the minimum accuracy requirement of 0.01 pH units (equivalent to the minimum pCO2 accuracy of 10 µatm for potential future inclusion into the SOCAT database) is not systematically achieved in the upper ocean."
- I also miss text on discrete samples of pCO2, DIC and TA (and their limitations)

Line 164: 'namely TA and pCO2' and 'DIC throughout the perturbed volume': please spare a few more words to explain, this is not obvious from the previous sentence.

Line 166: 'a carbon budget could theoretically be closed': add evidence or reference? (okay you say theoretically, but still)

Line 184: "restricted to the upper ocean (50m)": please check. I recently read that a SAMI-2 CO2 sensor has a maximum depth of 600 m, and CONTROS HydroC CO2 sensor has a maximum depth of 1000 m.

Line 184: "this is potentially important": does "this" refer to the depth restriction or to CO2 sensors being effective?

Line 186: "measurements of pCO2": according to the title, the Wanninkhof paper is about discrete measurements of pCO2, whereas this paragraph is about autonomous sensors. Are accuracies identical?

Line 188: "little involvement": isn't this a bit too optimistic, what about calibration? How long can they be deployed? A bit more explanation would be useful, as this is a central part of the manuscript.

Line 210: "fit-for-purpose models are not available": even if there were a reference to the Fennel et al chapter (which is not), this statement needs some more explanation, what are the major limitations?

Line 216: "ensemble": what kind of ensemble do you refer to? Perturbed parameters or perturbed initial conditions or different models?

Line 221: "quantifying uncertainty": how does DA help with quantifying uncertainty?

Line 222: Can you find a better title for this section? It is not about adding alkalinity to models.

Line 225: "air sea CO2 exchange": I'm missing "because…"

Line 228-230: please explain/elaborate and add references. You might also need to introduce and explain the term 'near-field'.

Line 230-233, 246-247: add references

Line 234: biological responses will have many more impacts than just on 'efficacy'. Expand and add refs.

Line 253-254: "further empirical research, "this aspect": can you be more specific, what precisely is missing? What do you recommend? What kind of research do we need? What about the inability of the models to accurately represent the background CaCO3 cycle (because we lack a good understanding from field observations; Planchat et al., 2023; Hinrichs et al., 2023)

Section 4.3: I don't understand what this section is about. Why would one neglect feedbacks of biology on circulation? This is included in many models anyway (see e.g. Seferian et al., 2020). It probably doesn't matter much (as you discuss), but in principle this should be included in the OAE simulation (and not in the CTRL/counterfactual simulation). What sort of models does this relate to?

Line 266: Oschlies reference missing in reference list.

Line 268-269: explain your expectation or add a reference.

Section 5:
- It is not quite clear why some recommendations are given as bullet points and others in plain text. Are they qualitatively different (then please explain)?
- I kind of expected a clear recommendation on what measurements to take, or at least that two carbonate system variables should be monitored continuously. I think in the Schulz et al chapter, there is a statement that pH and pCO2 together have higher uncertainty than any other pair of carbonate system variables. Would be useful to refer to this (it's a pity that they are the only ones with autonomous sensors available). If optimal monitoring is still to be figured out, state this.
- What about the overestimation of atmospheric CO2 reduction from ocean-centric MRV alone (see above)?

Line 277-279: add references (at least to Fennel et al.). Data-assimilation wasn't really discussed in this manuscript, it comes a bit as a surprise that this is a key conclusion.

Line 288-289: similarly, 'ecosystem effects and sediment-water exchange' were not discussed in this ms, why is it a key conclusion? If it stays, please elaborate.

Line 301-305: add references. (what is meant with "chapter 2"?)

Line 307: "should be done" → should ALSO be done

Line 315-317: this list of measurements is similar to what I expected as a major recommendation. The recommendation is well hidden though, and not all (not even most) of these measurements are explained in the manuscript.

Line 317: "measurements of carbonate chemistry parameters": too unspecific. How many? Which ones? Isn't this so central to OAE MRV that it should be mentioned more prominently?

Line 318ff: references needed

Line 326: "validated": not even evaluated

Line 329-330: well, the models exist and simulations are being produced. What is the problem? Is it not economically feasible for MRV to run them on supercomputers, or are not enough supercomputers or ... please specify.

Line 333: "establish that models": how? More/other computers? More efficient code? Missing processes? Please specify your recommendations.

Line 344: which observations? With regard to what? Please specify

Line 350: Shouldn't the company of the lead author be mentioned as competing interest?

Technical comments:

Line 14: "reporting so the" → reported

Line 106: OGCM → OGCMs

Line 118: mCDR was not introduced

Line 133: ""time scale" → time to reach equilibrium

Line 134: "Revelle buffer factor" → Revelle factor (buffer factor has the opposite behaviour, pick one)

Line 251ff: feedback → feedbacks

Line 315: ADCPs: spell out

References:

Köhler, P. (2023). Atmospheric CO2 concentration based on boron isotopes versus simulations of the global carboncycle during the Plio-Pleistocene. Paleoceanography and Paleoclimatology, 38, e2022PA004439. https://doi.org/10.1029/2022PA004439

Planchat et al., 2023: https://doi.org/10.5194/bg-20-1195-2023

Seferian et al., 2020: https://doi.org/10.1007/s40641-020-00160-0

Smetacek et al 2012, doi:10.1038/nature11229

Wimart-Rousseau, C., Steinhoff, T., Klein, B., Bittig, H., and Körtzinger, A.: Technical note: Enhancement of float-pH data quality control methods: A study case in the Subpolar Northwestern Atlantic region, Biogeosciences Discuss. [preprint], https://doi.org/10.5194/bg-2023-76, in review, 2023

---

## Author Response (AR1)

Response to public comments

From Steve Rackley

I congratulate the authors on producing a useful building block towards a widely accepted MRV methodology for OAE.

1) Line 204; a clarification and reference for the statement regarding coastal vs open ocean gas exchange velocity will be helpful. Wanninkhof's 2009 and 2014 papers do not support this differentiating statement, which is apparently based on areally limited coastal systems, specifically those above sea grass meadows.

*Response: We have added a reference.*

2) Line 244; the initial CDR resulting from OAE is determined by ocean conditions at the point (x,y,t) of air-sea equilibration, not by prior conditions, although credible history matching will obviously be important to establish model credibility. Ultimate CDR effectiveness will surely depend on the ocean conditions at the point (x,y,z,t) where an alkalized and equilibrated water parcel become isolated by subduction from contact with the mixed layer for all climate relevant timescales?

*Response: This chapter is about MRV. Models are used to compare the OAE deployment to the counterfactual baseline, which is why knowledge of the initial state is needed.*

3) Line 71 &c; The impact of biocalcification response on OAE effectiveness is an interesting and complex question. Whether it has any real impact on OAE effectiveness will depend on the assumed long-term counterfactual for ocean conditions.

*Response: Yes.*

4) Line 351; CarbonDirect and [C]Worthy seem worthy of mention here?

*Response: We have updated our affiliations.*

Response to reviewers

Reviewer 1 — Christopher R. Pearce

This manuscript by Ho et al. provides a useful overview of the requirements, current capabilities and future research priorities for facilitating effective Monitoring Reporting and Verification (MRV) of Carbon Dioxide Removal (CDR) by Ocean Alkalinity Enhancement (OAE). The paper is part of the invited series on 'Guide to Best Practices in OAE research' thus does not cover all aspects relating to OAE MRV, and only focusses on MRV challenges associated with quantifying CDR. The paper is exceptionally well written and structured and provides a very good overview that will be accessible and beneficial to both non-experts and practitioners alike. I therefore have no hesitation in recommending the manuscript for publication, and make only a few suggestions for consideration.

I recognise that the authors have intentionally only focussed on MRV for CDR for the reasons specified in lines 34-38. However, I think that the manuscript (and potentially the series as a whole) would benefit from a more explicit discussion on the requirements and current approaches for conducting MRV of environment/ecosystem impacts associated with OAE. The two other papers within the series cited on line 34 do not adequately cover the assessment of OAE effects on ecosystems in their current form: Subhas et al discuss natural analogues and comment on potential MRV platforms and approaches for conducting in-situ observations, while Fennel et al discuss a range of modelling assumptions and limitations, but neither explicitly cover how environmental or ecosystem impacts will be assessed or monitored during/after OAE. It's possible that these ecosystem impacts will be covered elsewhere in the series (in which that paper should be cited here), but if not, this chapter would seem like the logical place to include a more expansive discussion or section on MRV environmental or ecological impacts ('eMRV'), given that in practise such assessments are likely to be conducted simultaneously with MRV for CDR ('cMRV'). For example; Section 3 could easily be expanded to include techniques for monitoring other biogeochemical, environmental or ecological changes that may arise from OAE application (e.g. nutrient fluxes, particulate loading, phytoplankton community structure etc), or a separate section added to expand on this. Combining both 'eMRV' and 'cMRV' challenges within this chapter would also widen its relevance to interested parties and may provide useful guidance or suggestions how both requirements could be conducted concurrently.

*Response: We have updated the pointers to the other chapters where ecosystem impacts are explicitly addressed, and pointed out that determining these impacts would require monitoring other biogeochemical, environmental, or ecological changes that may arise from OAE application (e.g. nutrient fluxes, particulate loading, and phytoplankton community structure).*

All sections within the manuscript provide useful and accurate summaries of the current state of the art of MRV for OAE. However, I think that Section 3 (observational techniques and limitations) could be expanded to include other relevant information and approaches. For example; the authors refer to the measurement of TA and pCO2 as routes for directly quantifying the extent of CDR (lines 179-190) but they do not discuss the potential to constrain

these perturbations in seawater chemistry by other means – namely by monitoring pH (for which highly precise autonomous marine sensors are commercially available) alongside one of the other parameters. The authors also do not discuss the potential or limitations of directly measuring CO2 uptake into the oceans via covariance techniques, and I think that it would also be useful to consider the potential and limitations of using remote sensing approaches to quantify air-sea CO2 fluxes and/or other effects such as coccolithophore blooms during large scale OAE treatments. Expanding Section 3 in this manner, and/or better emphasising our potential to use the inter-dependencies of the marine carbonate system to support in-situ MRV observations elsewhere in the manuscript, would be useful additions.

*Response: We have now significantly expanded section 3, e.g. explicitly discussing the use of pH measurements and remote sensing approaches for large-scale OAE treatments. We note that measurements are also discussed in Eisaman et al (2023) and in Fennel et al (2023; section 3.1 Observation types for validation).*

*Regarding the potential for direct measurements of $CO_2$ uptake, we note that measuring $CO_2$ flux via eddy covariance will not work under most conditions because the measurements will need to be made from a fixed location and the alkalinity-infused water will have advected and dispersed. However, the bigger issue is whether it is necessary to directly observe the additional air-sea $CO_2$ flux, or whether an approach in which the perturbed chemistry (observed as a change on TA, pH and/or $pCO_2$) is accepted as producing a change in flux, with some "risk assessment for incomplete equilibration" as in Bach et al. (2023). We believe the latter is what will be achievable and should suffice for MRV.*

*Even though quasi-Lagrangian autonomous vehicles such as Argo floats can approximately track deep water masses, estimation of changes in water masses "infused" with alkalinity would require highly accurate measurements of carbonate chemistry parameters. Unfortunately, as demonstrated by Wimart-Rouseau et al. (2023), pH sensors on profiling floats have relatively large uncertainties that may compromise their usefulness for MRV.*

Technical comments

Line 45 (& elsewhere). I strongly urge caution with use of the term 'leakage' in a CDR context. Whilst I fully agree that the potential loss, or return, of carbon removed via the CDR intervention has to be quantified, I personally think that this is better presented as determination of the true or net efficacy of the approach (and all factors that contribute to this), rather than quantification of the amount of leakage. This is because the term leakage is already used within the CCS sector to define the potential release of dissolved or gaseous CO2 from geological reservoirs (in T/d) and may also be used to refer to the gradual release of waste or contaminants, neither of which are of relevance here.

*Response: We agree that the term "leakage" has ambiguity and have removed it.*

Line 112. The title of Section 2 seems to be miss-ordered. Something like "Specificities for MRV of CDR by OAE" would be a more appropriate (albeit acronym heavy) description.

*Response: We agree, and have changed the title to "Specificities of MRV for Ocean CDR"*

Lines 339-344. The authors refer to an OAE signal dispersal on a timescale of weeks to months before which the signals would be diluted and be harder to observe (line 343). Whilst this is likely to be the case for single alkalinity applications to a patch of open seawater, the scenario and response would differ significantly for repeated or continuous release in a both open or coastal settings. Consequently, although I agree with the point being made regarding the need to cross-validate MRV modelling approaches with observations, I do think this section should be rephrased to accommodate different OAE approaches.

*Response: We have rephrased the manuscript to include the idea that continuous application does not share this challenge to the same degree: "Unless alkalinity is continuously applied at a level measurable by long-duration observing platforms, the OAE signals are likely to be diluted and less easily tracked with observations."*

References. Some of the citations are missing journal information (e.g. Anderson et al. 2007 and Bach et al. 2023) and it would be useful to include the citations of the other papers within this series in the reference list.

*Response: We have added the missing journal information.*

Reviewer 2 — Anonymous

Review of Ho et al. : Chapter 6: Monitoring, Reporting and Verification for Ocean Alkalinity Enhancement

This paper is a contribution to a "Guide to Best Practices in OAE research", which is a timely and highly relevant endeavour. The chapter itself is rather short, maybe because it is sandwiched between the Fennel et al modelling chapter and the Schulz et al. chapter on carbonate chemistry measurements. Overall, the chapter is well readable and informative.

Besides a few specific comments, my major comment is on incomplete referencing. The full first section ("1. What is MRV") only includes one reference (plus two references to other chapters in this issue, which are, however, missing in the reference list). Even though I agree with most statements, the full chapter reads more like an opinion piece rather than a scientifically sound paper deeply rooted in the scientific literature. Given that the key message by the authors is that "early stage MRV research for OAE [...] carries a special obligation toward comprehensiveness, reproducibility, and transparency" (Abstract and section 5), I urge the authors to follow their own recommendation and to thoroughly reference the relevant literature. The links and references may be obvious to those deeply involved in this research, but this series of chapters should have the ambition to be a starting point to those new in the field. Thus, please point the reader to relevant literature, so that he/she/it can find further reading material. I give a few examples below, but this holds for the entire manuscript.
My first thought was that there might be a common decision/recommendation of a maximum number of references to be included in the chapters of the best practise guide. If this was the case, I would strongly urge the entire group to rethink their decision. However, looking at the Fennel et al. paper with a substantially longer (though maybe also not complete) reference list, this does not seem to be the case.

*Response: We have added several references to this section to support claims that have been substantiated in the literature. In some cases, however, the text is indeed communicating statements that are the primary content of this paper. In these cases, we have modified the language to make this clear.*

In the same spirit, the manuscript would benefit from sometimes sparring a few more words for an additional explanation when statements are rather vague and unspecific, examples are given below.

Specific comments:

Line 20: 'pCO2 and pH' : pH is not mentioned at all in section 3 on measurements. Please add.

*Response: We added a paragraph to describe current capabilities to measure pH on floats, gliders and moorings.*

Line 28-29 (and in general): MRV is defined to deliver 'the amount of additional CO2 removed from the atmosphere' and 'the durability of that removal'.

What I am missing in this paper is twofold:

any ocean-centric MRV being through observations or modelling would overestimate the atmospheric CO2 draw-down by neglecting feedbacks from atmosphere and particularly land and thus overestimating atmospheric CO2 reduction by ~2s% on decadal scales (Oschlies, 2009, www.biogeosciences.net/6/1603/2009/ ). There may be ways to account for this even if not using an emission-driven Earth System Model, but these approaches need to be developed, and awareness needs to be raised. I think this is mentioned in the Fennel et al chapter, but don't count on everyone reading all chapters.

*Response: We concur that converting carbon removal into atmospheric $CO_2$ drawdown necessitates the consideration of carbon cycle feedbacks, as demonstrated by Oschlies in 2009. However, it's important to note that this is also true for all other carbon dioxide removal (CDR) techniques and for emission reduction measures as well. These feedback mechanisms are indeed taken into account when determining carbon emission trajectories required to achieve global warming targets and, consequently, in the potential requirement for CDR in terms of carbon mass units. Therefore, we think that specifying these feedbacks in this context is unnecessary.*

 I think it is too simplistic to say that all carbon in the ocean counts equally as all alkalinity stays in the ocean for long time-scales. Isn't it still better (for humans) to get the carbon out of the surface mixed layer and ideally into water masses that will not be in contact with the atmosphere for centuries? I.e., what about the depth of carbon sequestration? Shouldn't this be included in MRV where the carbon goes (horizontally and vertically)?

*Response: The reviewer is right that the depth of carbon sequestration is another key metric for estimating the efficiency of ocean CDR (see Siegel et al. 2020) when atmospheric $CO_2$ is stored in the oceans as dissolved $CO_2$ (i.e., without alkalinity enhancement, which applies for example for seaweed CDR or Iron Fertilization). However, in the case of OAE, $CO_2$ is stored mainly as $HCO_3^-$, which cannot be exchanged with the atmosphere so surface ocean storage is chemically safe. Keeping alkalinity (and thus $HCO_3^-$) in the surface ocean has benefits for ocean acidification, although these are very minor (see "equilibrated OAE" in Fig. 3 in Bach et al., 2019). Furthermore, retaining alkalinity ($HCO_3^-$) in the surface can limit interactions with sediments and thus substantial loss terms to carbon removed via OAE such as the risk of inducing secondary $CaCO_3$ precipitation in sediments and the reduction of natural alkalinity release (Moras et al., 2022; Bach, 2023). We acknowledge that there are also loss terms to alkalinity ($HCO_3^-$) in the surface ocean, such as the induction of biotic calcification. However, due to the arguments above, there is currently no reason to believe the deep ocean is a much safer place to store atmospheric. $CO_2$ as $HCO_3^-$. We have added this explanation in Section 1 under the language about durability.*

Line 36: you here promise the reader some text on methane, N2O and DMS. I did not find this mentioned again... add text (preferred) or delete this in our outline.

*Response: Changes in radiative forcing by non-$CO_2$ climate-forcing agents have been shown to be of potential importance for other ocean CDR techniques (e.g. Dutreuil et al. 2009 with $N_2O$ and DMS for artificial upwellings). But there is so far no reason to think that OAE would lead to emissions of non-$CO_2$ climate-forcing agents through direct mechanisms - an exception being mineral-based OAE with iron contamination and OIF as a side effect. We have hence removed the mentions of methane, $N_2O$, and DMS.*

Line 45: "CO2 that escapes removal": there will be a lot of CO2 that will not be removed, so should this be reformulated to be more specific?

*Response: this text has been removed.*

Line 47: "such as": just wondering whether there is anything more to add to the list or if it is complete then drop "such as".

*Response: Since we are listing two examples, we feel that the use of "such as" is appropriate.*

Line 52: The long time-scale of alkalinity cycling in the ocean of Middleburg was challenged by Kohler (2023), who found, for example, a glacial-interglacial amplitude in alkalinity of 100 μmol/kg, which, according to the paper, is equal to 100 ppm in atmospheric CO2. This may not be relevant to the timescales of CDR, but some caution/toning down might be worth considering. I am not sure we have a sufficiently good understanding of changes in river/erosion input of organic and inorganic matter to the coastal ocean and neither of how sedimentary processes in coastal regions change with climate change and other human impacts to be so certain that alkalinity cannot be lost (other than through secondary precipitation and calcification).

*Response: We have revised the text to clarify that the residence time of alkalinity in the ocean far exceeds 10^3 years.*

Line 64: "stored as CaCO3": CaCO3 formation is usually thought to increase ocean pCO2, so the formulation "CO2 stored as CaCO3" seems to be misleading, and might need a few extra words.

*Response: We removed this statement because (although chemically correct), storage durability of $HCO_3^-$ is ~$10^5$ years and thus essentially already "permanent". A conversion of $HCO_3^-$ into $CaCO_3$ can increase durability to >$10^6$ years when alkalinity loss is compensated by non-carbonate weathering (e.g. silicate weathering). But we think this is more of an academic debate, rather than a topic that is relevant for carbon market requirements with regards to MRV.*

Lines 64-70, 73-74, 76-78, 81, 85-86, 91-93!, 95-100, 114-116, 137: essentially, each

sentence needs a reference.

*Response: References have been added.*

Line 82: one does wonder: there are a couple of global pCO2-products estimating the ocean carbon sink from pCO2-observations. How does MRV relate to these estimates? Do you also consider them unreliable or what makes the difference to quantifying air-sea CO2 flux after OAE from observations? Is it the small scales? It would be welcome, if you commented on that for the reader.

*Response:  This is a good point. There are sophisticated tools for estimating surface $pCO_2$ using existing observations and machine-learning techniques, with high-quality products emerging from neural networks that use predictor variables provided by satellites (e.g. sea surface temperature, chlorophyll) and/or from data-assimilating models (e.g. mixed layer depth) (e.g. Landschutzer et al., 2022).  However, their resolution (typically ~1° lon x lat and monthly mean) may be too coarse for many MRV needs, such as for comparisons to OAE deployments over small spatial scales or short time scales.  Additionally, some studies have shown that global products have relatively large errors in regions with high gradients (Nickford et al., 2022) and in near-coastal regions (Sharp et al., 2022), with RMSE averaging about 10 µatm, and much higher near the coast. Nevertheless, the global products have proved extremely valuable in studies of ocean carbon uptake and should be supported and enhanced with the collection of high-quality surface ocean $pCO_2$ observations. Machine learning techniques similar to those used to create global products can be trained for smaller regional studies, providing more highly-resolved, gap-filled products (e.g. Sharp et al., 2022). The resulting predicted $pCO_2$ from a neural network trained with data before any OAE deployment may be useful as a counterfactual against which to compare field data collected after the OAE experiment. We incorporated these thoughts as described in response to the next comment.*

Line 89: hmm, if a long-term climatology from observations would be used as a 'counterfactual scenario', what if the climate state/weather would be so different in the year of OAE (climate variability) that the different climate state/weather can explain all the differences in CO2 uptake. Should a company get credit for this?

*Response: Related to the previous response, we agree that it could make much more sense to compare the environment after the CDR perturbation to a counterfactual provided by a spatially- and temporally-varying product that is trained from historical data. Instead of relying on the deployment year being similar to the climatological mean year (a bad assumption), the (better) assumption is that the relationship between $pCO_2$ and observable variables like SST, chlorophyll, and MLD would have been similar in the deployment year as the years in which the training data was collected. The potential use case for these empirical products to create the counterfactuals underscores the value of maintaining and enhancing them, especially creating more locally relevant, highly resolved products that can be run by relevant groups in a timely manner. For the introduction, we now revise the statement in question to reflect that a*

*counterfactual could be provided from these statistical reconstructions and do not suggest using a simple climatology.*

Section "2: Specificities of ocean CDR for MRV": it would be nice to embed this for the reader in how this differs from land MRV (just a thought).

*Response: Agreed - how ocean CDR differs from land CDR, and subsequent implications for MRV, is a key point of this section. After introducing the specific time- and spatial scales of the ocean carbon cycle, we now mention that these time scales differ from the ones involved in the land carbon cycle, and hence add constraints on the design of MRV for ocean CDR.*

Line 122: 'coastal stations, where OAE is likely to be deployed': needs a ref or explanation

*Response: We have added that OAE is likely to be deployed in coastal areas because of proximity to existing infrastructure, energy supply, and human resources.*

Line 134: "ionization fraction": is this not a very similar measure to the Revelle factor?

*Response: It is not. The ionization factor is the ratio between DIC and dissolved $CO_2$ concentration. See Jones et al. (2014) for a detailed discussion on how this factor partly controls the time scales of air-sea $CO_2$ equilibrium. We will add a brief explanation of the difference in the revised manuscript.*

Line 136ff: not only alkalinity, also carbon will be transported (is that included in your 'OAE signal'?)

*Response: The sentence has been changed accordingly.*

Line 142ff: "Lessons learned from OIF": it would be useful to extend this part and explain better. Currently, I do not take any lesson home from this text.

*Response: This section has been changed to make the lessons from OIF more relevant to OAE.*

Line 147: "in the context of M (Measurement)": I thought M is Monitoring, but also I don't get the meaning of this phrase.

*Response: The mention of Measurement has been removed.*

Line 156-158: not true, at least not always, see Smetacek et al 2012: "A large diatom bloom peaked in the fourth week after fertilization. This was followed by mass mortality of several diatom species that formed rapidly sinking, mucilaginous aggregates of entangled cells and chains."

*Response: We have clarified the text to state that we are discussing artifacts done in oceanic waters unbounded by eddies. The Smetacek study was conducted in a large eddy.* Running trials within eddies can cause additional issues (not discussed here) such as different background signatures in the eddy center versus the periphery.

Line 159: I'm not sure these are 'experimental artefacts' and 'do not represent the ocean C cycle', but yes, even well-defined experiments take place in a complex Earth System and have limitations. The experiments took place in the real world though and do represent the carbon cycle, though maybe not in its entirety or the aspects that you implicitly refer to (please specify).

*Response: The text has been modified to state that such experiments with artifacts do not represent some key functions of the ocean carbon cycle, such as enhanced POC downward flux.*

Line 160: please explain why OAE would be less impacted by physical effects. Alkalinity and carbon are subject to circulation and mixing, only they may not be subject to the same potentially large amount of biological feedbacks.

*Response: We have clarified this sentence to point out that it is the influence of the physics on the biota.*

Section 3: Observation-based techniques:
as the abstract mentions pH, some text on pH sensors and their limitations should be added. I warmly recommend Wimart-Rousseau et al (BGD): "In the context of converting surface ocean pH measurements into pCO2 data for the purpose to derive air-sea CO2 fluxes, we conclude that the minimum accuracy requirement of 0.01 pH units (equivalent to the minimum pCO2 accuracy of 10 µatm for potential future inclusion into the SOCAT database) is not systematically achieved in the upper ocean."
I also miss text on discrete samples of pCO2, DIC and TA (and their limitations)

*Response: We are grateful to the reviewer for bringing this reference to our attention and we have included it in a new paragraph on pH sensing:*
Sensors that measure pH on autonomous profiling floats, gliders, or on moored platforms could provide additional data useful for MRV. Unfortunately, as demonstrated by Wimart-Rouseau et al. (2023), pH sensors on profiling floats have relatively large uncertainties that may compromise their usefulness for MRV. Moreover, these uncertainties are largest near the ocean's surface, where they would be most useful in the MRV context where the surface ocean disequilibrium is needed for CDR. Uncertainties in pH of 0.01 roughly translate to a $pCO_2$ uncertainty of 10 µatm (Wimart-Rouseau et al. (2023), but achieving such accuracy in pH will require significant advances in sensor accuracy and/or post-processing data analysis tools to correct surface pH data.

Line 164: 'namely TA and pCO2' and 'DIC throughout the perturbed volume': please spare a few more words to explain, this is not obvious from the previous sentence.

*Response: We've changed this to the following: Measurements of total alkalinity (TA) and DIC are important to quantify the background state of the carbon system, which determines the pCO2 response per unit change in alkalinity. Further, measurements of TA might help verify that alkalinity has been added effectively, although signal-to-noise ratios may be insufficiently strong to enable robust detection and attribution of TA anomalies (Mu et al., 2023)*

Line 166: 'a carbon budget could theoretically be closed': add evidence or reference? (okay you say theoretically, but still)

*Response: Changed to the following text: With extensive measurements of these variables along the Lagrangian pathway of a perturbed water mass, a carbon budget could theoretically be closed by constraining the time-rate of change and making inferences about important driving processes such as air-sea exchange; such a budget could in theory be used to support quantification of and CDR quantified for a given OAE deployment.*

Line 184: "restricted to the upper ocean (50m)": please check. I recently read that a SAMI-2 CO2 sensor has a maximum depth of 600 m, and CONTROS HydroC CO2 sensor has a maximum depth of 1000 m.

*Response: We have removed this part of the sentence.*

Line 184: "this is potentially important": does "this" refer to the depth restriction or to CO2 sensors being effective?

*Response: We have clarified that "this" refers to the existence of autonomous $pCO_2$ sensors.*

Line 186: "measurements of pCO2": according to the title, the Wanninkhof paper is about discrete measurements of pCO2, whereas this paragraph is about autonomous sensors. Are accuracies identical?

*Response: The equilibrator-based autonomous instruments could be as good as the lab-based measurements. The membrane-based ones have other challenges that degrade their accuracy.*

Line 188: "little involvement": isn't this a bit too optimistic, what about calibration? How long can they be deployed? A bit more explanation would be useful, as this is a central part of the manuscript.

*Response: This has been revised to:* These sensors have the advantage of being able to collect measurements continuously in harsh weather with much reduced involvement from skilled analysts relative to field surveys with bottle collection. Most analysis focuses on collecting and analyzing calibration samples and performing quality control on sensor data.

Line 210: "fit-for-purpose models are not available": even if there were a reference to the Fennel et al chapter (which is not), this statement needs some more explanation, what are the major limitations?

*Response: We removed "fit-for-purpose" and changed the sentence to highlight that "developing and validating" models for OAE should be a priority. We now include a reference to the modeling chapter.*

Line 216: "ensemble": what kind of ensemble do you refer to? Perturbed parameters or perturbed initial conditions or different models?

*Response: We added the following text:*
*Notably, there are different potential sources of uncertainty: if intrinsic variability in ocean dynamics is considered the dominant source of uncertainty, an initial condition ensemble could provide an appropriate representation of uncertainty. If model structure, in contrast, is the dominant source of uncertainty, alternative approaches to ensemble construction could be employed, including perturbing parameters or using multiple models (see Chapter [Modeling-chapter] for further discussion).*

Line 221: "quantifying uncertainty": how does DA help with quantifying uncertainty?

*Response: We think DA can help quantify one source of uncertainty when models with and without DA are compared. However, here, we have deleted the mention of DA helping with quantifying uncertainty.*

Line 222: Can you find a better title for this section? It is not about adding alkalinity to models.

*Response: The section title has been changed to "Modelling alkalinity addition".*

Line 225: "air sea CO2 exchange": I'm missing "because..."

*Response: We have modified the text to specify potential time scales for alkalinity additions and for air-sea $CO_2$ equilibrium.*

Line 228-230: please explain/elaborate and add references. You might also need to introduce and explain the term 'near-field'.

*Response: Some examples of near-field processes are discussed in the sentences that follow the section highlighted by the reviewer. We have added some references and elaborated that near-field means "within a few km of the source".*

Line 230-233, 246-247: add references

*Response: Done*

Line 234: biological responses will have many more impacts than just on 'efficacy'. Expand and add refs.

*Response: This is discussed in the next section with a reference to Fennel et al. (2023).*

Line 253-254: "further empirical research, "this aspect": can you be more specific, what precisely is missing? What do you recommend? What kind of research do we need? What about the inability of the models to accurately represent the background CaCO3 cycle (because we lack a good understanding from field observations; Planchat et al., 2023; Hinrichs et al., 2023)

*Response: The need of empirical research for improving models is further developed in Fennel et al. (2023, this volume). This reference has been added.*

Section 4.3: I don't understand what this section is about. Why would one neglect feedbacks of biology on circulation? This is included in many models anyway (see e.g. Seferian et al., 2020). It probably doesn't matter much (as you discuss), but in principle this should be included in the OAE simulation (and not in the CTRL/counterfactual simulation). What sort of models does this relate to?

*Response: We have deleted this section.*

Line 266: Oschlies reference missing in reference list.

*Response: This section has been deleted.*

Line 268-269: explain your expectation or add a reference.

*Response: This section has been deleted.*

Section 5:
It is not quite clear why some recommendations are given as bullet points and others in plain text. Are they qualitatively different (then please explain)?

*Response: The bulleted list provides overarching, almost philosophical recommendations. The text in the paragraphs is more specific advice for measuring and modeling efforts related to MRV. The distinction has been noted in the text.*

I kind of expected a clear recommendation on what measurements to take, or at least that two carbonate system variables should be monitored continuously. I think in the Schulz et al chapter, there is a statement that pH and pCO2 together have higher uncertainty than any other pair of carbonate system variables. Would be useful to refer to this (it's a pity that they are the

only ones with autonomous sensors available). If optimal monitoring is still to be figured out, state this.

*Response: Optimal monitoring has, indeed, not yet been defined, and will likely be deployment-specific until some standardization can be established based on early field trials. Until we know more about how MRV might be optimized, we default to the broad argument for being as comprehensive as possible. Once we have extremely complete - even redundant - observations of early field trials, protocols can be updated with more parsimonious approaches. We try to communicate this more clearly in the revised text.*

-       What about the overestimation of atmospheric CO2 reduction from ocean-centric MRV alone (see above)?

*Response: See the response above*

Line 277-279: add references (at least to Fennel et al.). Data-assimilation wasn't really discussed in this manuscript, it comes a bit as a surprise that this is a key conclusion.

*Response: Reference to Fennel et al. (2023) added.*

Line 288-289: similarly, 'ecosystem effects and sediment-water exchange' were not discussed in this ms, why is it a key conclusion? If it stays, please elaborate.

*Response: This part is in Fennel et al. (2023) and we provided a reference to it.*

Line 301-305: add references. (what is meant with "chapter 2"?)

*Response: chapter 2 was a reference to the part of the OAE guide on carbonate chemistry by Schulz et al. (2023). We have updated this reference accordingly and added a reference to Eisaman et al. (2023).*

Line 307: "should be done" ➔ should ALSO be done

*Response: The sentence was changed accordingly.*

Line 315-317: this list of measurements is similar to what I expected as a major recommendation. The recommendation is well hidden though, and not all (not even most) of these measurements are explained in the manuscript.

*Response: We have rewritten this as a bulleted list*

Line 317: "measurements of carbonate chemistry parameters": too unspecific. How many? Which ones? Isn't this so central to OAE MRV that it should be mentioned more prominently?

*Response: We have specified that these should be parameters that should change due to OAE, meaning TA, pH, and pCO$_2$.*

Line 318ff: references needed

*Response: We have added some references.*

Line 326: "validated": not even evaluated

*Response: "validated" changed to "evaluated".*

Line 329-330: well, the models exist and simulations are being produced. What is the problem? Is it not economically feasible for MRV to run them on supercomputers, or are not enough supercomputers or ... please specify.

*Response: The reviewer is right. This is now clarified in the revised text.*

Line 333: "establish that models": how? More/other computers? More efficient code? Missing processes? Please specify your recommendations.

*Response: We added a sentence to clarify this section*

Line 344: which observations? With regard to what? Please specify

*Response: We have specified that the model results should be consistent with the carbonate chemistry and tracer observations.*

Line 350: Shouldn't the company of the lead author be mentioned as competing interest?

*Response: We have added verbiage about DH and ML's role in [C]Worthy, LLC, a non-profit research organization focused on building tools to support MRV for ocean CDR.*

Technical comments:

Line 14: "reporting so the" ➔ reported

Line 106: OGCM ➔ OGCMs

Line 118: mCDR was not introduced

Line 133: ""time scale" ➔ time to reach equilibrium

Line 134: "Revelle buffer factor" ➔ Revelle factor (buffer factor has the opposite behaviour, pick one)

Line 251ff: feedback ➔ feedbacks

Line 315: ADCPs: spell out

*Response: Fixed*

References:

Kohler, P. (2023). Atmospheric CO2 concentration based on boron isotopes versus simulations of the global carboncycle during the Plio-Pleistocene.
Paleoceanography and Paleoclimatology, 38, e2022PA004439.
https://doi.org/10.1029/2022PA004439

Planchat et al., 2023: https://doi.org/10.5194/bg-20-1195-2023 Seferian et al., 2020: https://doi.org/10.1007/s40641-020-00160-0 Smetacek et al 2012, doi:10.1038/nature11229 Wimart-Rousseau, C., Steinhoff, T., Klein, B., Bittig, H., and Kortzinger, A.: Technical note: Enhancement of float-pH data quality control methods: A study case in the Subpolar Northwestern Atlantic region, Biogeosciences Discuss. [preprint], https://doi.org/10.5194/bg-2023-76. in review, 2023

Reviewer 3 — Anonymous

The authors have presented an overview of the current status of monitoring, reporting and verification (MRV) as it relates to a (marine) Carbon Dioxide Removal (mCDR) activity. They have outlined the current state of research for MRV with respect to Ocean Alkalinity Enhancement (OAE) in particular, as part of an invited series on Best Practices for OAE Research.

The paper is well written and comprehensive, and will in my opinion, suitable for publication as part of this collection with minor revisions. Some additional comments for the authors to consider are listed below.

Define DIC before first use (line 51)

*Response: Done*

Define pCO2 before first use (Line 73)

*Response: Done*

Line 80 – long time (not 'long-time')

*Response: Fixed*

Line 85 – suggest avoiding the introduction of a new term (air-to-sea transfer of CO2) and consistently use air-sea CO2 flux, or exchange.

*Response: Fixed*

Line 95 – geographic patterns for CO2 flux are also controlled by biological processes

*Response: Added*

Line 97 – this sentence seems to link the annual cycle of heating with phytoplankton conversion of inorganic carbon to organic carbon, which is misleading.  Suggest rephrasing this and the above to include biology as a driver of spatial variability in air-sea exchange, then introduce the thermal component in the following text.

*Response: We have clarified this sentence.*

Lines 95-99 – seems like many of these statements could use citations.

*Response: We have added references.*

Line 105 – remove " from tracers.

*Response: Fixed*

Fennel 2023 reference is missing from the text, without reading that paper it is difficult to assess whether a reference to it is sufficient to cover all aspects of OBMs in the context of MRV for OAE.

*Response: Reference added*

Line 113 – reference(s) needed for 'typically spanning more than 10 orders of magnitude'

*Response: Reference added*

Line 125 – it would be useful to include some values for context here – e.g., an addition of x umol/kg alkalinity may be expected to induce a change in pCO2 of x uatm.

*Response: This is a great suggestion. We have added the following sentence: "For example, in the best case scenario, an increase in alkalinity of 10 µmol/kg would result in a decrease in pCO2 of around 20 µatm (given temp =20°C; salinity = 35; initial TA = 2200 µmol/kg; DIC = 1965 µmol/kg)*

Line 145 – 154 – these lines seem to assume that the OAE will take place in the open ocean, while logistically, at least in early trials, it is likely that these will be coastal efforts. Perhaps a few lines dedicated to the expected issues of subduction or spreading in a coastal context could be added here? Also not clear what the discussion of OIF adds here, seems to distract from the issue of time scales of relevance for OAE?

*Response: We do present an example of how the subduction of an SF6 labeled OIF patch resulted in the premature termination of the experiment. We have added a sentence to address subduction in a coastal context for OAE.*

*The OIF experiments were done at unprecedented scales (100-1000 km2) and so provided some key lessons for OAE. such as the alteration of the initial perturbed patch by ocean physics and the growth of the perturbed patch driven by ocean physics. We have added 2 more sentences to better link the lessons for OIF to that of OAE.*

Line 162 – not clear why suddenly 'total' is added before alkalinity.

*Response: Removed*

Line 163 and elsewhere – consistent with pCO2 or pCO2

*Response: Fixed*

Line 170 – as this is intended as a best practice guide, it would be useful to comment on the detection limits of the parameters being discussed – e.g. 5 umol/kg for alkalinity sensors, 2 umol/kg for discrete samples.

*Response: Relevant details about the detection limits for different sensors (two flavors of pCO2, pH, and TA) have been added. For example, for the PIC sensor: the detection limit is expected to be 0.5 microgram PIC per liter (with a range up to 500 microgram per liter which corresponds to a coccolithophore bloom).*

Line 172 – doesn't seem that tracking each 'molecule' is what is being discussed, but rather each 'patch' of alkaline water

*Response: Since carbon accounting requires knowing the amount of $CO_2$ removed, tracking each molecule of $CO_2$ is just an extreme case of that accounting.*

Line 185 – include reference to Sutton et al., for USVs (Sutton, A. J., N. L. Williams and B. Tilbrook (2021). "Constraining Southern Ocean CO2 Flux Uncertainty Using Uncrewed Surface Vehicle Observations." Geophysical Research Letters 48(3):e2020GL091748.)

*Response: Added*

Line 190 and above – commercially available pH sensors should be discussed here in terms of detection limits and utility in baseline observations of the CO2 system as well as in tracing an alkaline plume of water (along with pCO2, for example).

*Response: We added the following: Sensors that measure pH on autonomous profiling floats, gliders, or moored platforms could provide additional data useful for MRV. Unfortunately, as demonstrated by Wimart-Rouseau et al. (2023), pH sensors on profiling floats have relatively large uncertainties that may compromise their usefulness for MRV. Moreover, these uncertainties are largest near the ocean's surface, where they would be most useful in the MRV context to deduce the pCO2 deficit needed for CDR. Uncertainties in pH of 0.01 roughly translate to a $pCO_2$ uncertainty of 10 μuatm (Wimart-Rouseau et al. (2023), but achieving such accuracy in pH will require significant advances in sensor accuracy and/or post-processing data analysis tools to correct surface pH data.*

Line 203 – the section on windspeed seems somehow out of place, not clear that there will be a need for dual tracer experiments in coastal locations to meet MRV requirements.

*Response: We have added additional context for the need for wind speed measurements and also tracer measurements.*

Line 280 – might be worth citing the Planetary MRV as an example of 'open' sharing of information? (https://github.com/Planetary-Technologies/MRV)

*Response: Added*

Line 295 – not clear why the natural variability should only be known in the case of adding minerals via pulverised rock – seems that this would be a requirement for any source of alkalinity prior to addition?

*Response: We have changed the requirement to all techniques.*

Reviewer 4 — Justin Ries

The authors have done an admirable job of reviewing and discussing some of the key factors involved in MRV of OAE-based CDR.

That said, the manuscript seems somewhat overly focused on quantifying the transfer of $CO_2$ from the atmosphere to ocean through the highly variable and difficult to measure flux of $CO_2$ across the air-sea interface, when the most important flux to quantify is the more general transfer of $CO_2$ from shorter residence time reservoirs (including atmospheric $CO_2$ or aqueous $CO_2$) to longer residence time reservoirs (including the dissolved bicarbonate or carbonate ion reservoirs). In support of this concept, the authors acknowledge that net $CO_2$ removal from the atmosphere can be induced by OAE even by reducing the outgassing of $CO_2$ from the ocean without physically transferring $CO_2$ from the atmosphere to the ocean. This is a critical point to make in this best practices guide to OAE MRV, but one that is not sufficiently addressed in the present version of the submission. Focusing on the highly variable, difficult to measure, and unnecessary air-to-sea $CO_2$ flux is somewhat misleading and creates artificial obstacles for a technology that will likely be needed as one of the core CDR pathways to even approach IPCC targets by the end of this century.

*Response: We don't agree with the framing that the transfer of carbon from shorter residence time reservoirs like aqueous $CO_2$ to longer residence time reservoirs like bicarbonate is central to OAE. For example, if this transfer of aqueous $CO_2$ to bicarbonate happens at depth — where seawater will not be in contact with the atmosphere for long periods — it will have no impact on atmospheric $CO_2$, which is the goal of OAE deployments.*

*Furthermore, we are not saying that we need to measure air-sea $CO_2$ fluxes following the delivery of alkalinity. This would be impossible since the time and spatial scales are so large. This is why MRV has to be based on trustworthy models, which are calibrated and evaluated by data. Furthermore, there are practical solutions to deal with our inability to directly measure air-sea $CO_2$ fluxes after an OAE deployment, which we now mentioned in this contribution (e.g. Bach et al., 2023). In this reference, we also explicitly addressed how obstacles can be reduced.*

*With regards to influx/outflux, we mentioned them more explicitly in the revised version.*

I would also recommend that the authors consider incorporating into the introduction a brief overview of the carbonate chemistry equilibria in seawater and how OAE impacts these equilibria to transfer $CO_2$ from the shorter residence time atmosphere and aqueous $CO_2$ reservoirs to the longer residence time bicarbonate and carbonate ion reservoirs – hence constituting durable CDR.

The referencing could also be improved in places as noted below, especially with respect to quantitative statements about carbonate system dynamics in seawater.

*Response: A detailed description of carbonate chemistry, central to the understanding of OAE, is provided in Schulz et al. (2023), in this same issue.*

All of that said, this is a complicated subject and the present submission represents the first attempt of which I am aware to establish a framework for quantification of mCDR by OAE, and the authors should be commended for their efforts on this.

Line 20: wouldn't measuring elevations in DIC be a more reliable measure of OAE-based CDR than ocean pCO2 or pH?

*Response: In a model, where an OAE deployment could be evaluated against a control (counterfactual), changes in DIC is a good metric to evaluate the success of an OAE deployment. However, with respect to measurements, because the change in DIC is so small relative to the inventory of DIC in the ocean, it is better to measure $pCO_2$ or pH, as we elucidate later in the contribution.*

26: considering replacing 'MRV research' with 'OAE research'

*Response: We changed this to "MRV for OAE".*

28: It seems that the concept of marine CDR should be expanded from 'transfer of CO2 directly from the atmosphere into seawater' to 'transfer of CO2 in the atmosphere or seawater into stable carbonate or bicarbonate ions in seawater', as both processes (i.e., transfer of CO2 from atmosphere and seawater) will result in the eventual drawdown of CO2 from the atmosphere. Otherwise, we will miss an important and efficient pathway in CO2 removal. Likewise, CO2 removal from the atmosphere alone is not sufficient for CDR, as increasing the pCO2 of the atmosphere through increased CO2 emissions also increases the flux of CO2 from the atmosphere to the ocean, but surely this should not constitute marine CDR. This point is also illustrated by the authors' mention of the scenario by which OAE lessens CO2 release to the atmosphere (rather than increasing flux of CO2 from the atmosphere to the oceans, which would still lead to a net reduction in atmospheric CO2 yet would not satisfy the authors' current requirement that OAE CDR be based on the direct measurement of CO2 flux from the atmosphere to the ocean. Perhaps a more useful framing for CDR is the transfer of C from shorter residence time reservoirs (atmospheric CO2, seawater CO2, terrestrial biomass, marine biomass in mixed layer etc.) to longer residence time reservoirs (bicarbonate ion reservoir, carbonate ion reservoir, terrestrial and marine biomass transported to deep ocean below mixed layer, etc.) (c.f., Prentice, I. C.,

2001, The carbon cycle and atmospheric carbon dioxide. Climate change 2001: the scientific basis, Intergovernmental panel on climate change. hal-03333974) or, more colloquially, transferring C from the 'fast C cycle' to the 'slow C cycle', as this encompasses the ultimate goal of marine CDR – i.e., net reduction of atmospheric $CO_2$, regardless of the strict, difficult, and not necessarily relevant measurement of $CO_2$ flux across the air-sea interface.

*Response: We addressed this point about the transfer of carbon from the short-term to long-term reservoirs above.*

*In the OAE guide, of which this contribution is a chapter, CDR is defined as "Human activities that lead to a net removal of $CO_2$ from the atmosphere, either by increasing the removal of $CO_2$ from the atmosphere or by reducing the emissions of $CO_2$ from natural sources to the atmosphere, and that durably store the removed carbon away from the atmosphere."*

30: consider adding 'ocean acidification' to 'climate change' as a reason for marine CDR

*Response: This sentence is about the motivation for investment in CDR. Millions of dollars are not pouring in to mitigate ocean acidification (otherwise this would have happened long ago). Furthermore, it's true that OAE would initially increase pH at the point of deployment. However, if OAE were fully efficient, the amount of atmospheric $CO_2$ invading the ocean would negate the initial pH increase.*

31: 'accurate estimates' is a somewhat ambiguous term; consider changing to 'MRV must provide estimates of net carbon removal and the uncertainty of those estimates'

*Response: The sentence has been modified accordingly.*

45: change to 'amount of CO2 over time'

*Response: This section has been eliminated in response to another reviewer's comment*

60: consider clarifying that quantifying dissolution of added alkalinity is also not a concern for alkalinity added in a dissolved form (i.e., as an alkaline brine), in addition to the already mentioned production of alkalinity through electrolysis.

*Response: This sentence has been changed to mention this point.*

61: consider changing 'mitigating' to 'reversing'

*Response: "Mitigating" has been replaced by "decreasing the efficacy"*

63: 'carbonate minerals'

*Response: Done.*

66: citation needed for assertion that calcification increases exponentially with saturation state; I agree that dissolution rate increases exponentially below omega of 1 (as expected from the universal dissolution rate equation), but most prior studies I am familiar with show calcification rates increase relatively linearly once omega gets much above 1.

*Response: Agreed. 'Exponentially' has been removed.*

68: because the manuscript deals with carbonate ions as well as calcium carbonate, the term 'carbonate' should be avoided for clarity and replaced with the two aforementioned terms as appropriate

*Response: Agreed. These terms have been changed accordingly.*

71: should clarify that $MgCO_3$ does not precipitate from seawater under normal (non-OAE) conditions

*Response: We do not feel that this clarification is relevant in this sentence, where we are addressing OAE.*

73: clarify that OAE generates a reduction in 'seawater $pCO_2$', rather than just '$pCO_2$'

*Response: Done*

74: add 'due to Henry's Law'

*Response: We do not feel that this adds extra information to the sentence.*

75, 80: quantifying air-sea flux of $CO_2$ in response to OAE is not feasible for the reasons stated by the authors (too slow and small to measure with readily available autonomous instrumentation due to dilution and slow kinetics of air-sea equilibration). It is also not necessary for MRV because the critical step is the transfer of aqueous $CO_2$ to the bicarbonate or carbonate ion reservoirs. This is a key issue that should be clarified in the manuscript.

*Response: We addressed this point about the transfer of carbon from the short-term to long-term reservoirs above.*

85: air-to-sea CO2 transfer, in isolation, is not necessarily required to be known (or really that reliable) to quantify and verify mCDR by OAE for the reasons discussed above; changes in DIC and total alkalinity (or pH and one of these parameters, with the other parameter calculated) provide a more complete picture of the transfer of CO2 into the long-term HCO3- and CO32- reservoirs, which are the most critical pathways to quantify for verifying CDR by OAE

*Response: We addressed this point about the transfer of carbon from the short-term to long-term reservoirs above.*

95: again, in the case of OAE sequestering CO2 on a net basis by reducing outgassing of CO2, this change in air-to-sea CO2 flux does not need to be measured, and it would be nearly impossible to do so in the open ocean. Measurement of changes in DIC and total alkalinity (or pH and one of these parameters, with the other parameter calculated) should provide a more complete understanding of the impact of OAE on the carbon cycle and, ultimately, net changes in atmospheric pCO2.

*Response: We addressed this point above.*

97: these variations also exist in the tropics, such as on coral reefs, when photosynthesis by coral zooxanthellae can cause variations in pCO2 between 300 and 1200 uatm.

*Response: We have added biology as one of the causes of $pCO_2$ variations.*

113: this statement would benefit from greater specificity; e.g., which aspect of the ocean C cycle varies by 10 orders or magnitude – fluxes, reservoirs, etc.

*Response: This sentence is elucidated by the next sentence: "These scales range from that of the ocean skin, a thin layer of less than a millimeter in contact with the atmosphere where air-sea $CO_2$ exchange is controlled by molecular diffusion, to that of the global ocean circulation that typically transports dissolved carbon over more than a thousand years and 10,000 km."*

117: delete 'very' to avoid editorialization

*Response: Deleted.*

124: specify what is meant by 'variability' here; range? standard deviation? Also, specify what are the +/- units on the variability (range, standard deviation, etc?)

*Response: We have indicated that this is inter-station mean and standard deviation*

125: but the effect of OAE will be on top of the systematically fluctuating (season, diurnal) baseline values of TA and pCO2, so just because the effect of OAE on TA and pCO2 is less than the natural variability does not mean that this measuring these parameters will not reveal net transfer of CO2 to longer residence time reservoirs of C – the primary criterion for CDR.

*Response: We addressed this point about the transfer of carbon from the short-term to long-term reservoirs above.*

127: 'carbonate equilibria' instead of 'equilibrium'

*Response: replaced*

133: consider changing 'initial' to 'baseline'

*Response: replaced*

155: this section would benefit from elaboration on how OIF transforms the patch into a chemostat, especially for the non-specialist reader

*Response: The language about chemostats has been removed in response to another reviewer's comment.*

184: Citation needed for measurement of pCO2 with accuracy of 2 uatm

*Response: Done*

184-185: Although the authors state that seawater pCO2 can be measured to 2 uatm accuracy, what really matters in terms of the error propagation and accuracy of using measured parameters (pCO2, pH, TA, DIC) to calculate carbonate system parameters is the number of significant figures in the measurement (i.e., number of decimal places when expressed in scientific notation). For example, if pCO2 can be measured to uatm accuracy, then a measured pCO2 of $4.51 \times 10^2$ uatm would contain 2 significant figures and thus limit the number of significant figures in calculated carbonate system parameters to 2 significant figures if combined by measurements of TA or DIC or spectrophotometric pH that can typically be measured to 3 significant figures (i.e., $2.125 \times 10^3$ umoles/kg-sw, spec pH = $8.132 \times 10^0$). However, if TA and/or spec pH and/or DIC are used to calculate carbonate system parameters, which can be measured to more significant figures than pCO2 (2 significant figs), then the remaining carbonate system parameters can be calculated with greater accuracy.

*Response: Measurements for MRV of OAE would be during the initial perturbation, where we would want to see changes in TA and pCO$_2$ for model input, calibration, and evaluation. While a*

*discussion about how accurately one could calculate different carbonate system parameters given the uncertainties in each parameter, it is not relevant to this section of the MRV chapter.*

193: though intuitive, a citation needed for assertion that coccolithophores may proliferate under high alkalinity and pH conditions

*Response: Reference to Bach et al. (2019) has been added.*

195: delete 'intrinsic'; add 'and aragonite' as presumably this would be required to quantify PIC (e,g., aragonitic pteropods, etc) as well.

*Response: Thank you for pointing out this omission. Rephrased as "Autonomous optical sensors for particulate inorganic carbon (PIC) based on the birefringence of calcite and aragonite have also been in development for several decades by Bishop et al. (2009, 2022)."*

286: it looks like this is the first time that 'mineral ballasting' has been mentioned (unless I missed an earlier mention) and it would benefit from a more thorough discussion (or at least a description) either here and/or earlier in the manuscript.

*Response: Good point; we have added a paragraph on mineral ballasting in section 3.*

300: again, terms like 'the precise amount of' are somewhat ambiguous; if precision is going to be mentioned as a goal, then the levels of precision sought should be described. Alternatively, reframe quantification of precision as the basis for the uncertainty discount, rather than requiring or implying that some given (arbitrary) level of precision must be achieved.

*Response: We have rephrased in the manuscript to "If the enhancement is done via electrochemistry, the dosing rate of the solution (e.g., Mg(OH)$_2$, NaOH) should be quantified and reported with complete information about the measurement methods and a thorough accounting of their uncertainties"*

306: the term 'drawdown of pCO2 in the ocean' seems to be a non sequitur; shouldn't it state 'transport of CO2 to the ocean' or 'drawdown of atmospheric pCO2'. Also, as discussed above, transfer of CO2 from atmosphere to ocean is not necessarily the parameter that matters most, as net CO2 removal from the atmosphere can be induced by OAE even by reducing the outgassing of CO2 from the ocean without physically transferring CO2 from the atmosphere to the ocean. This is a critical distinction to make in this best practices guide to OAE MRV but one that is not sufficiently addressed in the present version of the submission. It is the transfer of CO2 from short residence time reservoirs like atmospheric or aqueous CO2 to longer residence time reservoirs (e.g., bicarbonate and

carbonate ion reservoirs) that provides the most comprehensive picture of what is happening to the C cycle and, ultimately, atmospheric pCO2.

*Response: "Drawdown of pCO$_2$" changed to "drawdown of CO$_2$". We addressed this point about the transfer of carbon from the short-term to long-term reservoirs above.*

323: hyphenation not needed for adverb modifiers ('empirically determined')

*Response: hyphen removed*

Misc:

Confirm that 'ocean alkalinity enhancement' and 'carbon dioxide removal' should be capitalized

*Response: We have removed the capitalization.*

Intro would benefit from a brief overview of the carbonate chemistry equilibria in seawater and how OAE impacts these equilibria to transfer CO2 from the shorter residence time atmosphere and aqueous CO2 reservoirs to the longer residence time bicarbonate and carbonate ion reservoirs – hence constituting durable CDR

*Response: A detailed description of carbonate chemistry, central to the understanding of OAE, is provided in Schulz et al. (2023), in this same issue.*

Respectfully submitted,

J. Ries

---

## Author Response (AR2)

Many thanks David and co-authors! This is all great, the reviewer comments have been addressed adequately. Based on my own reading, I still have a few minor points I would like to raise and ask you to address in the final version:

(i) you state that "it is difficult to detect changes in the carbon inventory of the ocean with measurements of DIC" (p.8 first line) and never provide a good explanation. At the same time you argue that the added alkalinity should and can be measured to some extent. The signal-to-noise ratio of additional alkalinity and additional DIC should be very similar. Why is measuring DIC not given a more positive consideration? Could you please explain or rephrase the possible use of DIC measurements?

It is true that TA and DIC will have similar signal-to-noise after a few months to years in the ocean flow. However, we meant that TA increase should be measured in the near field after deployment, and that it would be impossible to use an increase in DIC in the far field to determine CDR.

We have added the following sentences to the section that discusses alkalinity:

It is worth noting that measuring a TA increase near the OAE deployment point may be possible, but once the OAE-perturbed water has dispersed in the ocean flow, the signal-to-noise ratio will likely be too low to make any accurate quantification. This is also the case for attempting to quantify CDR using DIC, as discussed below.

(ii) the citation of other papers of the Guide should be "Fennel et al. (2023, this Guide)" etc.

Fixed

(iii) the DA discussion in chapter 4, page 9, bottom: In ocean-only models, the counterfactual condition could use the same physics and the same initial conditions for ocean biogeochemistry as the DA solution for the OAE condition, but for the counterfactual just run without OAE. This neglects possible (but usually unlikely) direct effects of OAE on the ocean physics. It assumes that DA produces dynamically consistent trajectories (Ensemble filters or adjoint), which would be required for mass conservation anyway, with mass conservation likely being very useful if not mandatory for models used in MRV. If this is correct, wouldn't the counterfactual problem be much smaller than you indicate?

This is true if you are only assimilating physical variables. However, if you are assimilating BGC variables, there is still a problem.

Sorry for bringing this up only now. Happy to learn that my concerns are not valid. If they are, they should be straightforward to address via a few changes in the text.

Thanks and best wishes,
-Andreas